# The Use of Cognition by Amphibians Confronting Environmental Change: Examples from the Behavioral Ecology of Crawfish Frogs (*Rana areolata*)

**DOI:** 10.3390/ani15050736

**Published:** 2025-03-04

**Authors:** Michael J. Lannoo, Rochelle M. Stiles

**Affiliations:** 1Department of Anatomy, Cell Biology, and Physiology, Indiana University School of Medicine, Rm 135 Holmstedt Hall-ISU, Terre Haute, IN 47809, USA; 2San Francisco Zoological Society, 1 Zoo Road, San Francisco, CA 94132, USA; rochelles@sfzoo.org

**Keywords:** amphibian declines, noetic knowledge, activity patterns, breeding status, ecological disturbance, ontogeny, interoception, exteroception

## Abstract

One overlooked advantage amphibians possess in the struggle for survival is their brains share the same blueprint as human brains, which allows them to acquire knowledge and understanding through experiences—in other words, amphibians have cognitive capabilities that assist them in their effort to survive. Here, we use four examples from our work on the behavioral ecology of Crawfish Frogs (*Rana areolata*) to hypothesize how cognition affects amphibian reactions to environmental and social change. We offer that as one component of our fight to conserve amphibians, researchers should consider the full range of anatomical, physiological, and behavioral features amphibians themselves employ in their defense, which are features responsible for their historical evolutionary success up until the Anthropocene.

## 1. Introduction

Organisms are not billiard balls, propelled by simple and measurable external forces to predictable new positions on life’s pool table. Sufficiently complex systems have greater richness.—Stephen Jay Gould [1] (p. 16)

We say of other creatures, “Ah, they’re just animals,” and they are. But we have to expand our definition of animal every time we get to know one better.—Douglas H. Chadwick [2]

Our primary interest is in the behavior of the living body, and we study brains because these organs are the chief instruments which regulate behavior.—C.J. Herrick [3] (p. 5)

One casualty of the recent emphasis on the ‘global’ decline of amphibian populations is the concept that amphibians have historically been survivors. Since they evolved ~365 million years ago [4,5], amphibians in three separate orders (Anura, Caudata, and Gymnophiona) have endured innumerable planetary and geologic catastrophes, as well as climate extremes. There can be no doubt amphibians have historically been a successful group. Today, there are roughly 8800 recognized species of amphibians [6] versus 6750 recognized species of mammals [7].

Are amphibians declining? Yes, they most certainly are [8,9,10,11,12]. Some of the most evolutionarily novel amphibians on the planet have been driven to extinction [6,13,14] by factors such as habitat loss, climate change, and disease [8,11], with each loss being a tragedy. Are amphibians declining everywhere? No, they are not. While no species can survive the obliteration of its habitat, or ecological conditions that exceed the maxima or minima of its collective individuals, this has not yet been the situation in most ecosystems or regions across Earth.

The standard view of amphibian declines presumes they are passive victims of anthropogenic environmental change—Gould’s [1] billiard balls being knocked around by anthropocentric factors beyond their control—and if humans do not intervene, most amphibians will succumb [15]. But this belief ignores the physiological, biochemical, and behavioral flexibility amphibians can employ in the face of inevitable environmental challenges [16,17,18,19,20,21].

One overlooked advantage amphibians possess in the struggle for survival is that, being vertebrates, their brains are organized into regions dedicated to specific sensory systems and motor tasks as well as areas, especially in the forebrain diencephalon and telencephalon, dedicated to integrating the information that is received and coordinating an appropriate response. The brains of anurans resemble squalomorph sharks in that most neurons are unmigrated (i.e., remain close to the ventricle [22]; this is especially true in tadpoles [23]), but anuran brains also contain a substantial number of neurons that are migrated. In contrast, the brains of salamanders are much less differentiated than salamanders through an evolutionary process termed neoteny, or the retention of juvenile characters into adulthood [3,24].

Amphibian brains share the same blueprint as human brains [3,22,25,26,27]. Notably, except for relatively minor variations, for example, in visual system projections, cerebellar circuitry, and upper motor innervation [28,29,30], the brainstem and cranial nerves of adult frogs are so like that of humans in both their arrangement and function that they have been used as models in medical school anatomy labs (Figure 1). While many early neuroanatomists were reluctant to accept the idea that subcortical nuclei found in the human brain are present in and function similarly to the brains of more basal vertebrates (but see [3]), today, such acceptance is the standard [22,31,32]. We will be referring to this principal as ‘homologous nuclei have homologous functions’.

Where amphibian and human brains differ is in the forebrain telencephalon; mammals have evolved a cerebral cortex—an overlayer of neuronal processing that refines and supplants more ancestral circuits [22,26,33,34,35]. In addition to receiving olfactory inputs, the human cerebrum is responsible for generating conscious solutions to problems, as well as creating rationalizations, language, emotions, drives, and dreams. To perform these functions, the human cortex uses cognition, or the mental process of acquiring knowledge and understanding through sensations, perceptions, and experiences (involving memory and learning [36]). Among vertebrates, many forms of cognition have been recognized including animal or comparative cognition (to distinguish from human cognition), physical cognition (knowledge of the world), and social cognition (knowledge of one’s own species [37]). For a review of amphibian cognitive abilities, see Table 1 in [38]. Vertebrate cognition in all its forms elaborates on a discontinuum broadly correlated with the expansion of the forebrain telencephalon, culminating in the human cerebral cortex [27,34,39,40,41].

As a useful construct, Fabbro and colleagues [35] divide the vertebrate brain into a basal system and the telencephalon. The basal system includes the spinal cord, brainstem (hindbrain plus midbrain), and the forebrain diencephalic thalamus and hypothalamus, and is responsible for ‘anoetic’, or reactive knowledge. In contrast, the telencephalon is responsible for ‘noetic’, or knowing knowledge. Here, we will be referring to noetic knowledge in amphibians as cognition, based on the following observations.

In the nineteenth century, the eminent philosopher/psychologist William James conducted a series of observations on frogs (species not stated, likely *Rana temporaria*) following the ablation of various portions of their central nervous system [42] (see Figure 1). When James severed the brains of adult frogs between the telencephalon and diencephalon, he noted the casual observer would consider the frog to be normal. However, missing its cerebrum, these frogs produced no spontaneous motion, lacked the motivation to eat, and showed no fear. James describes these frogs as “a machine… By applying the right sensory stimulus [to the animal]… we are almost as certain of getting a fixed response as an organist is of hearing a certain note when he pulls out a certain stop” [42]. In contrast, James observed that normal frogs exhibited spontaneous and complex acts of locomotion “as if moved by what in ourselves we should call an idea”. Further, he wrote, they made “persistent and varied efforts at escape, as if the notion of danger… were [their] spur”. When hungry, James states these frogs searched for prey such as insects, fish, and smaller frogs, and altered their attack with each type of prey. In sum, James felt when a frog’s telencephalon was intact, its behavior became variable and unpredictable, demonstrating it could learn and was aware of itself and its environment—that it was using cognition (Figure 2).

James’s observations [42] neatly fit modern notions of knowledge [35]. Amphibians who have had their telencephalon ablated and are responding to their environment using only diencephalic, brainstem, and spinal cord circuitry are using anoetic or reactive knowledge, while amphibians with intact telencephalons demonstrate noetic knowledge and again can be said to possess cognition. Amphibians also have a ‘cognitive perspective’ which means they can infer another animal’s intent [35]. This is critical information when that other animal is a potential mate or approaching predator.

We lack evidence that amphibians possess ‘autonoetic’ knowledge—experiencing the world from the perspective of future hopes and aspirations. Nor do we have evidence amphibians have an ‘affective perspective’ which is the ability to make inferences based on other organisms’ emotions and feelings (i.e., while amphibians mate, the attraction is temporary and functional; they do not form life-long bonds; but see [43]). Nor do we consider every pathway within or function of the telencephalon to contribute to cognition. For example, the olfactory vomeronasal system is responsible for subconscious pheromone detection [44,45] but see [46,47,48].

Tests probing the extent of amphibian abilities have traditionally used controlled or manipulated observations in the laboratory [34,49,50,51,52,53,54]. For example, in a comparative study between a social, diurnal tropical frog (*Dendrobates auratus*) and a more solitary, nocturnal frog (*Engystomops pustulosus*), Burmeister has shown *D. auratus* (the social species) has a greater degree of gene expression associated with neurogenesis in the ventromedial pallium (the hippocampus, the region thought to be responsible for memory and learning), suggestive of a tight link between neuronal capacity and social/ecological demands [55].

Our goal here, suggestive of Burmeister’s approach [38], is to follow Edelman and colleagues’ plea [56] to use alternate strategies for assessing consciousness by expanding the catalog of examples of amphibian cognition into the natural world of ecology—amphibians in the wild interacting with their biotic and abiotic environment (i.e., cognitive ecology [57,58,59]). Specifically, we ask in what situations do amphibians use cognition to respond to ecological and social challenges? We are not claiming amphibians consciously think their way through their problems. We are simply challenging any notion of amphibians as billiard balls being passively knocked around life’s felt surface by the cue stick of climate change. This is a concept that fails to acknowledge the benefits that amphibian telencephalic circuitry provide, which are advantages this highly successful basal tetrapod group have employed in response to the challenges they have faced in the 365,000 or so millennia of their existence. We submit that as one component of our fight to conserve amphibians, we should be appreciating the full range of anatomical, physiological, biochemical, and behavioral features amphibians themselves deploy in their defense; these features were responsible for their evolutionary success up until the Anthropocene. To explore this perspective, we use four examples from the behavioral ecology of Crawfish Frogs (*Rana areolata*), one of the behaviorally and ecologically best-known species of North American amphibians [60]. We retrospectively examined 15 years of data, field experiences, and peer-reviewed publications on this Indiana State Endangered and Globally Near-threatened species (https://www.iucnredlist.org, accessed on 10 December 2024) to consider whether cognition drives amphibian responses to environmental challenges. With this understanding of their natural history, Crawfish Frogs offer a model to inform future research on cognition in amphibians. Moreover, if current threats to their existence [60] persist, it is imperative that we learn as much as we can about how these frogs interact with their world before all populations are extirpated.

## 2. The Basic Behavioral Ecology of Crawfish Frogs

Crawfish Frogs have been called the most secretive true frog in North America [61]. When not breeding in seasonal or semipermanent wetlands Crawfish Frogs occupy abandoned burrows dug by crayfish, the origin of their common name (Figure 3). Burrows can be a meter or more deep, ending in a broad chamber that under normal hydric conditions holds water. Each crayfish burrow usually supports only one frog. What makes Crawfish Frogs unique is they typically occupy a single crayfish burrow, what we have called their primary burrow, for the entire eleven or more months when not engaged in breeding activities [62,63]. Remarkably, primary burrows can be located more than a kilometer away from breeding wetlands [62,63], 10,000 times the length of a 100 mm SVL frog.

Crawfish Frogs emerge from their primary burrow to feed, which they do on a “feeding platform” adjacent to the burrow entrance. When on its feeding platform, a Crawfish Frog faces the burrow entrance and when frightened, dives into its burrow, turns around, inflates its body against the burrow walls, and lowers its head. Burrow fidelity centers around the fit of frog to burrow. Frogs cannot enter burrows too small for them, and if occupying a burrow too large are unable to wedge their bodies against its walls to create resistance. Once a Crawfish Frog finds a burrow that fits their body, they adopt it. We have documented Crawfish Frogs using the same primary burrow for up to five consecutive years [60].

When a Crawfish Frog is in its burrow facing up towards the burrow entrance, it uses its head to protect its soft body parts. To facilitate this strategy, the jaws of Crawfish Frogs grow allometrically [64] such that when seen from above they form a semicircle, allowing their jawline to fit the burrow curvature and seat into the burrow passage, something like a lid on a bucket [60] (Figure 3). This is a remarkable and effective defensive strategy when facing a predator such as a snake or weasel that can enter the burrow. However, being tied to a primary burrow makes Crawfish Frogs especially vulnerable to agricultural disking and plowing, which is the reason why they are a species of conservation concern nearly everywhere they occur.

Our research on Crawfish Frogs was conducted between 2009 and 2016 at the Hillenbrand State Fish and Wildlife Area located in southwestern Indiana. To date, our work has produced 37 peer-reviewed papers, which we have recently compiled and put into context [60]. As mentioned above, here we use four examples from this work to infer how cognition affects how Crawfish Frogs respond to environmental and social change. The first two examples describe their responses to seasonality and reproductive status, the third details their reaction to ecological disturbance, and the fourth details how their response to the same stimulus changes with growth/age. In each example, we detail the neuronal circuitry thought to be involved and hypothesize the role if any of cognition.

## 3. Example 1: Shifts in Activity Patterns in Response to Seasonality

Most animals restrict their activity to a portion of the daily light cycle, being active either during daylight (diurnal), at night (nocturnal), or at dawn and dusk (crepuscular). Crawfish Frogs respond to the world differently, as evidenced by photographs taken by wildlife cameras we deployed on frog burrows from 2009 to 2013 (intervals between photographs ranged from 5 min to 1 h). Depending on the season, Crawfish Frogs are diurnal or nocturnal; they can also be circumdiel, active both diurnally and nocturnally—around the clock [65,66]—as follows (Figure 4):

—In the early spring and late fall, Crawfish Frogs are *diurnal*—active during the day when it is warm but in their burrows during chilly nights.—In the late spring and early fall, when both day and nighttime temperatures are moderate, Crawfish Frogs are *circumdiel*, active around the clock.—During hot, dry summer conditions, Crawfish Frogs avoid these desiccating seasonal conditions by becoming *nocturnal*.

Further, and perhaps more remarkably, this variable seasonal activity pattern was synchronized across all individuals—solitary adult males, adult females, and juveniles—within our study population (Figure 5).

This behavioral flexibility and its synchronicity are consistent with the interpretation that Crawfish Frogs are reacting to their environment in a deterministic, anoetic way, much like James’s machine [42], responding externally (behaviorally) to internal (physiological) temperature and moisture assessments tied to seasonal environmental variation. The circuitry underlying this behavior involves cutaneous receptors sending signals through spinal cord (body) and cranial nerve V, VII, IX, X (head) somatosensory pathways and their nuclei to the midbrain torus semicircularis (Figure 2). The torus then organizes these inputs then sends them upstream to the midbrain tectum and the forebrain diencephalic dorsal thalamus [29,67,68]; these brain regions together appear to be responsible for integrating sensory inputs. (Frogs with their tectal lobes lesioned have difficultly orienting to sensory stimuli [50,51,69,70,71]). Somatosensory information also reaches the diencephalic hypothalamus, which manages the animal’s homeostatic system, including water balance and thermoregulation, through both the hardwired autonomic nervous system and pituitary gland hormones (in particular, arginine vasotocin [72] circulating through the bloodstream [73,74,75,76,77,78]).

According to this circuitry and applying parsimony, we hypothesize the highest order of integration necessary to maintain temperature and moisture balance in amphibians occurs in the forebrain diencephalon (thalamus and hypothalamus), which interacts with the midbrain tectum to produce movement into and out of burrows. This interpretation is consistent with Tinbergen’s impression [79] of amphibians as “reflex machines” [34]. If this explanation holds, only anoetic knowledge is necessary to explain these seasonal variations in Crawfish Frog activity patterns—i.e., the telencephalon is not necessary and therefore its cognitive abilities need not be employed [35]. It is James’ parallel to hearing a particular note when an organist pulls out a certain stop [42]. This is analogous to, and perhaps in somatosensory and diencephalic circuitry homologous to, shivering in mammals in response to cold and sweating in humans in response to heat.

In short, while Crawfish Frogs may employ cognition in their daily activity pattern response to seasonal temperature and moisture variations (in particular, telencephalic-based basal ganglia circuits may be involved in the initiation and cessation of movements [80,81,82]; Figure 2 connections from A to MP and VM to Str), since their response and its synchronicity across solitary animals can be explained through basic spinal cord-brainstem stimulus-response circuits, we find no proof of cognition. In other words, Crawfish Frogs were nocturnal in the summer not because they remembered they were nocturnal during hot, dry periods in previous years but because they were avoiding the hot dry conditions they were currently experiencing [66].

## 4. Example 2: Vocal Transitions from a Non-Breeding to a Breeding State

Adult male Crawfish Frogs generate two kinds of vocalizations. The most common and well known is the breeding or advertisement call produced in a breeding wetland that serves to advertise the male’s presence and perhaps its status (bigger frogs produce lower frequency vocalizations [83], and male size enters into a female’s choice of mate [21,84]). In contrast, the much rarer upland call is produced terrestrially during non-breeding seasons at the frog’s burrow entrance. Upland calling may serve as a component of burrow defense and can be triggered by noise pollution [85] (Figure 6).

The anoetic interpretation of this behavioral pattern is ‘vocalize on land = upland call; vocalize in water = breeding call’. However, immediately prior to breeding, call production in Crawfish Frog males gets more complicated.

Anticipating the 2013 breeding season, we augmented the wildlife camera trained on Crawfish Frog 26’s burrow with a SongMeter audio recording unit (Wildlife Acoustics Inc., Concord, MA, USA) [85]. Both devices tracked time, allowing us to synchronize the audio to the camera images. During the evening of 11 March at 2354 h (sunset was 1951 h), with other Crawfish Frog males calling from a wetland about 100 m away, Frog 26 emerged from his burrow and began calling. He completed three calling bouts (Figure 7). Calls from the first bout were characteristic of upland calls. Calls from the second bout demonstrated alternating elements of upland-like and breeding-like calls. Calls from the third bout demonstrated characteristics intermediate between upland-like and breeding-like calls that graded into breeding-like calls. Following this third bout, Frog 26 migrated to his breeding wetland, where we captured him the next morning.

Given the production of these intermediate call types, it is clear Crawfish Frogs do not always follow the deterministic ‘terrestrial = upland calling, aquatic = breeding calling’ rule. To explain these transitional bouts, it is useful to consider the mechanisms underlying call production in male frogs.

Breeding call production in anurans is dependent on seasonally produced androgens circulating in the bloodstream [86,87,88]. Androgens stimulate a neuronal central pattern generator (CPG) located in the brainstem. This CPG consists of paired parabrachial nuclei (PbN) located in the rostral hindbrain wired to the motor nuclei of cranial nerves IX and X, which drive the vocal muscles [89]. While this hormonally driven circuit could explain the production of context-dependent upland and breeding calls [90], given the poor temporal resolution of hormonal influences, it is unlikely single hormones can, by themselves, account for the repeated 0.25 s shifts in the transitional call types (Figure 7) produced by Frog 26 prior to initiating his breeding migration. A different explanation is required.

In addition to hormonal influences on pacemaker circuitry, PbN and CN IX and X nuclei are hardwired to higher order nuclei, what Zornik and Kelley term “upstream vocal initiation centers” [88]. These centers include the serotonergic midbrain and hindbrain raphe nuclei, mesencephalic tectum (which, importantly here, receives auditory input from the midbrain torus semicircularis), diencephalic preoptic nuclei, and telencephalic central amygdala [29,91,92,93,94]. Knowing this circuitry, we can explain Frog 26’s behavior either in anoetic or noetic ways.

The anoetic hypothesis suggests each call type is driven by an environmental stimulus (location, conspecific calling) that releases hormones that act on different brainstem regions or nuclei. If stimuli exist for both call types simultaneously the behavioral output will produce unpredictable combinations of the two call types. The challenge to this hypothesis is its complexity is beyond what we understand brainstem circuitry acting alone can accomplish.

The noetic hypothesis is more aligned with James’s observations [42], as follows. Being at his burrow, Frog 26 included in each calling bout an appropriate upland-like call. But, primed by androgens and perhaps influenced by the proximity of nearby Crawfish Frog males producing breeding calls [95], he also included in each bout a breeding-like call. After completing these bouts, his drive to breed overcame his burrow fidelity and he began his breeding migration. In this interpretation, we hypothesize Frog 26’s transitional calling behavior was a product of his environment (being at his burrow), the drive to breed (seasonal hormone circulation), and social factors (hearing nearby calling males and perhaps promoting arousal and an endocrine response). Whether these transitional calls were produced by one pacemaker nucleus shifting states (the most likely scenario) or two different pacemakers competing [96] is unknown, but this complex series of behaviors must be the product of at least three separate sensory-motor systems (vocal production, hearing conspecifics, and locomotion to the source of calling conspecifics) and cannot be explained by what we know about reflexive, basal, anoetic stimulus-response circuits. Instead, Frog 26 exhibited situational knowledge (the transition from upland to breeding calls), species knowledge (attention and attraction to other calling male Crawfish Frogs), and self-knowledge (the drive to migrate to the source of conspecific breeding calls suggests a ‘them and me’ understanding). Under this hypothesis, the integration of these behaviors represents a cognitive circuitry likely originating in the telencephalon. Why? Because its complexity is beyond what we understand brainstem circuitry alone can accomplish. Note that while this transition from an upland state to a breeding state was non-linear in its details, it was linear in function in that it began with a male Crawfish Frog producing upland-like calls at its burrow and ended in that male producing breeding calls entering a breeding wetland.

In 2011, one of us (MJL) experienced another situation where cognition appears to have been involved in Crawfish Frog breeding call production. It was a cool night (<10 °C), at least a half-dozen males had migrated into the wetland where he was working, and the migration of breeding females had not yet started. A Crawfish Frog in a wetland 500 m away began calling, triggering a single male in the wetland MJL was working to begin calling. These frogs alternated calls over several chorusing events throughout the night [60], demonstrating they were aware of each other and reacting [97,98]. If this male had been the only male in the wetland MJL was working, we might call it reflexive. However, there were other males in this wetland that had called in previous nights. They must have heard the distant male’s call, were presumably influenced by similar levels of circulating androgens, yet did not call. While we expect variation in individual performance [99], we suggest this type of variation falls under the description ‘they chose not to call’, which, if true, is cognition, perhaps mediated in the telencephalon by the central amygdala in the circuitry described above, or by septal and striatal nuclei as detailed by Walkowiak and colleagues [100].

## 5. Example 3: Behavioral Changes in Response to Ecological Disturbance

On 16 August 2011, Indiana Department of Natural Resources property managers conducted a prescribed burn encompassing about 32 ha of our study site. This burn was hot and immolated most herbaceous vegetation down to mineral soil. At this time we had wildlife cameras set up on four Crawfish Frogs inhabiting burrows in vegetated prairie areas not burned. Recognizing an opportunity, following the burn we searched the scorched area and found four Crawfish Frog-occupied burrows (Crawfish Frogs easily survive prairie fires in the depths of their burrows) and set up cameras on these burrows as well. Our goal was to compare the behavior of frogs having herbaceous vegetative cover as protection from aerial and ground-based predation with frogs having no such advantage [101].

Our study lasted 6 weeks, from 26 September to 4 November. After analyzing over 24,500 photographs of the eight animals, we discovered while Crawfish Frogs occupying burned and vegetated habitats exhibited similar behaviors at night, their daytime activity differed dramatically (Figure 8). In daylight, compared with frogs in vegetated areas, Crawfish Frogs in the burned area spent much more time in their burrows (50.4% compared to 30.3%) and when outside their burrows spent more time at the burrow entrance (35.9% compared to 9.8%), usually with just their head protruding (Figure 9).

The anoetic interpretation of this behavioral shift could be that Crawfish Frogs in the burned area remained in their burrow entrances to avoid higher surface temperatures. During our study, temperatures at the burrow entrances ranged from 2 to 35 °C and were similar between the vegetated and burned areas in the evenings, at night, and during the mornings, but averaged 4 °C higher in the burned area in the afternoons, consistent with a heat-avoidance hypothesis. One fact, though, argues against this idea. Crawfish Frogs in the burned area emerged from their burrows later in the morning, as temperatures were heating up, and were active during the hottest part of the day. Rather than avoiding heat, they appeared to be seeking it. Given this observation, we reject the anoetic idea that Crawfish Frogs in the burned area were responding in a deterministic way to temperature. Instead, based on the behavioral pattern we observed we hypothesize Crawfish Frogs in the burn were ‘reluctant’ to show themselves. In the morning when frogs in the vegetated areas were emerging from their burrows and positioning themselves on their feeding platforms, frogs in the burned area appeared hesitant to emerge, and when they did, they tended to expose only their head (Figure 9).

The important difference between this example and the previous two is here we cannot tie behavioral differences to *actual* variations in environmental factors such as temperature, humidity, location, or conspecific advertisement calls. Instead, we hypothesize the reluctance to expose themselves exhibited by Crawfish Frogs in the burned area was a reaction to the *potential* threat of a visually driven predator occasionally present [102]. Adult Crawfish Frogs are solitary when not breeding and therefore do not have the opportunity to learn from observing the fate of others [103,104]. Rather, we hypothesize their aversion to exposure was either innate or a reaction to the memory of a previous encounter with a predator—what cognitive scientists call learned inhibition [38]. Crawfish Frogs, like other amphibians, form memories and can therefore learn [104,105,106,107,108,109]. In our study, Crawfish Frogs migrated from breeding wetlands back to distant burrows they had left days or weeks before, a feat they could accomplish only if they remembered (i.e., learned) the location of their burrow (see also [110]). By avoiding predation in the absence of a predator, Crawfish Frogs are not simply responding to an environmental or social stimulus. Instead, we hypothesize this behavior, based on self-awareness and perhaps memory (both a product of the telencephalon [38,55]), constitutes a form of logic and can rightfully be considered cognitive.

From an ecological perspective this is not a trivial behavioral shift. Burrows constrain mobility and since adult frogs orient their body towards a potential prey before lunging to engulf and ingest it [53,111], any restriction of movement likely reduces the number and success of feeding attempts. Tradeoffs between the probabilities of finding prey (reward) and becoming prey (risk) have drawn serious attention from behavioral ecologists [102,112,113].

## 6. Example 4: Ontogenetic Changes in the Perception of Other Organisms

Crawfish Frogs have a complicated relationship with Common Gartersnakes (*Thamnophis sirtalis*). Frog 460 (88 mm SVL) shared his burrow with a small (estimated 45 cm-long) Gartersnake, while Frog 401 (93 mm SVL) was eaten by a large (80 cm-long) Gartersnake. Further, if Crawfish Frogs have food habits like other large ranids [114], large adult Crawfish Frogs will occasionally eat small Gartersnakes.

Ewert demonstrated adult European Toads (*Bufo bufo*) discriminate visually between potential predators and prey using relative size and position [53]. Toads consider small moving objects at ground level as potential prey while large objects in the air are potential predators. This tenet is anoetic and resembles our Example 2 above where male Crawfish Frogs might tie rules about call type to their location (burrow = upland call, wetland = breeding call). But as we saw with these calling behaviors, there are circumstances in nature where simple rules break down.

So how do adult Crawfish Frogs consider Gartersnakes? Critically, because a mistake could mean death, we hypothesize Crawfish Frogs must assess the Gartersnake’s size relative to their own size. For a large adult frog, small snakes represent potential prey, medium-sized snakes can safely be ignored, and large snakes are predators. But Crawfish Frogs live several years before they become a large adult (their maximum longevity is at least 10 years [60]). To assess a snake based on its size, a Crawfish Frog must have some notion of its own size, and this notion must be continuously recalibrated as it grows. To a newly metamorphosed Crawfish Frog (~33 mm SVL, 4 gr) almost every snake no matter its size must be considered a predator. As the frog grows (adults in our studies averaged ~95 mm SVL and 95 gr), small- and medium-sized Gartersnakes can be ignored, while large snakes continue to be threats. As mentioned above, Frog 401 was eaten by an 80 cm-long Gartersnake. For older, larger Crawfish Frogs (110 mm SVL, 110 gr), all medium- and large-sized Gartersnakes can stop being viewed as predators and small snakes may begin to be considered prey. Again, for these shifts in perspective to occur, we hypothesize Crawfish Frogs must have a sense of their own absolute or relative size, and therefore possess an awareness of certain attributes contributing to their sense of self. Since we know of no deterministic hindbrain, midbrain, or diencephalic forebrain circuits that post-metamorphically switch states based on size or age, we hypothesize this knowledge must involve telencephalic circuits continuously uploading, storing, and downloading knowledge, which is cognition [36].

As defined by Fabbro and colleagues [35], we hypothesize Crawfish Frogs likely also have a cognitive perspective of a Gartersnake’s behavior. That is, as a Gartersnake approaches, Crawfish Frogs must infer intent. A Crawfish Frog detecting a Gartersnake of any appreciable size approaching quickly would be wise to dive in its burrow and assume a defensive posture. A Gartersnake approaching causally and attentive to other matters might deserve monitoring but nothing more.

## 7. Discussion

While Crawfish Frogs may not need cognition to respond to their physical- and chemical-based homeostatic needs such as temperature and moisture balance (Example 1), we hypothesize cognitive processes are necessary for their successful behavioral interactions with other animals, including conspecifics (Example 2), potential predators (Example 3), and animals that can be predators or prey depending on relational body sizes (Example 4).

Our hypotheses on the role cognition plays in the lives of amphibians rely on James’s observations [42], confirmed by Detwiler [115,116], of the role the amphibian telencephalon plays in creating behavioral novelty that rightfully can be called cognition [38,110,117]. Among the components of this form of animal cognition, we hypothesize Crawfish Frogs possess self-knowledge, ecological cognition (acquired knowledge of their biotic and abiotic world), social cognition (knowledge of their species), and probably cognitive knowledge in the sense they can infer intent in a potential mate or predator. These forms of cognition likely have been critical to the survival of amphibians since they first arose.

### 7.1. The Neuroanatomical Substrate

What features of the telencephalon might create cognition? The telencephalon is divided into a dorsally situated pallium and a ventrally situated subpallium [118] (Figure 1). The amphibian pallium is divided into medial, lateral, ventral, and perhaps dorsal nuclei [22,29,34,119]. The medial pallium is thought to be a homolog of the amniote hippocampus and subiculum—the structures that, based on the ‘homologous nuclei have homologous functions’ rule, upload long-term memories but probably do not store them [22,110,120]. Pallidal nuclei reciprocally connected with the medial pallium may serve to store memories [29,121,122].

The subpallium consists of a medial septum [123,124,125,126] and a lateral region containing the amygdaloid complex plus the basal ganglia [127,128]. The circuitry of the basal ganglia in anurans is essentially identical to its circuitry in all vertebrates [26,32,129] and consists of an input region called the striatum and an output region called the pallidum [32,80,81,82,130,131,132]. Inputs to the striatum are multimodal [80,133,134] (Figure 2). Outputs from the pallidum project to all thalamic and brainstem motor areas [81,82]. Through its ability to initiate and terminate motor patterns, the basal ganglia appear to be the highest-order brain region in amphibians controlling motor function (the extrapyramidal circuit [135]). Based on the ‘homologous nuclei have homologous functions’ rule, the basal ganglia of anurans control the ‘on and off’ aspects of complicated movements while the cerebellum plans movements and coordinates muscles as these movements are occurring.

In addition to the basal ganglia, the lateral complex of the ventral subpallium contains the amygdaloid complex. Located partially in the pallium, partially in the striatum, the amygdala receives olfactory inputs and is typically divided into several regions [77]. Collectively, these regions receive direct inputs from both the regular and vomeronasal olfactory systems, and indirect inputs from visual, auditory, somatosensory, and gustatory nuclei. Amygdaloid nuclei in turn send outputs to the hypothalamus where they affect autonomic, somatic, and endocrine responses [74,77,78].

Also located in the striatum and considered to be part of the outer shell of the amygdaloid complex, the dopaminergic nucleus accumbens traffics in goal- and reward-directed motor behavior [136]. The nucleus accumbens receives multimodal inputs from external and internal senses [80] and sends outputs to the amygdala, hypothalamus, and other forebrain and brainstem nuclei, including the serotonergic raphe nuclei [81].

The anuran amygdaloid complex, including the nucleus accumbens, plays an important role in the labeling of sensory stimuli as either beneficial (nucleus accumbens [136]) or dangerous (amygdala [135]). Through connections with the medial pallium, these labels are remembered and condition future behavioral responses to similar situations [38,55,77,117,137,138]. Again, the application of such acquired knowledge defines cognition.

### 7.2. How Do Amphibians Perceive the World Using This Circuitry?

Damasio and Damasio propose the origin of knowledge acquisition lies in internal, or interoceptive “homeostatic feelings” which “translate the process of life regulation and include salient fluctuations, e.g., hunger, pain, wellbeing, and states closer to equilibrium, e.g., plain feelings of life/existence” [139]. Perceptions of the outside world (i.e., interactions with biotic and abiotic aspects of an organism’s environment—i.e., its ecology) are meditated by externally oriented senses (vision, hearing, olfaction, touch, taste) and constitute exteroception. These authors suggest there is no interoceptive equivalent in exteroception: “we cannot have the map of a landscape interact with the landscape itself, which means [organisms and their environment] cannot commingle in the same way that they do in the world of interoception” [139]. We have less confidence in this statement. For example, the amphibian midbrain contains three-dimensional maps in the tectum (the superior colliculus of mammals) and torus semicircularis (the inferior colliculus of mammals) of an animal’s environment and projects onto them inputs from multiple sensory systems [22,67,68,140,141]. The fact that a frog will orient towards the location of a fly whether it sees it or hears it is an example of neural maps of a landscape (located in the midbrain) commingling with features of the landscape (the fly; see also [54]).

In fact, all animals live in and are dependent on their external environment, where they occupy ecological niches that serve to meet their interoceptive-driven homeostatic needs. When an animal’s interoceptive system signals hunger, that need is met through the organism’s exteroceptive knowledge of the landscape it inhabits. Feelings of thirst, or of being too hot or too cold, are met similarly by knowing in which ecosystems it can find water or shelter. If vertebrates can exhibit salient fluctuations in response to interoceptive inputs [139], they should also be able to exhibit salient fluctuations in response to the exteroceptive inputs serving these interoceptive needs, and this is where cognition becomes important.

### 7.3. Do Behavioral Ecologists Need to Know Details of Neural Circuitry?

Both behaviorists and ecologists have tended to follow Skinner [142] in viewing brains as black boxes we need not see into or understand [143]. For example, in a clever series of behavioral experiments designed to test memory retention in Wood Frogs (*Rana sylvatica*) tadpoles, Chivers and Ferrari refer to the telencephalic circuitry involved not as a neuroanatomical wiring diagram but instead as a “complex algorithm used by animals to acquire, encode, and use information from their environment” [108]. But it is not 1938 anymore. As daunting as this task of working through the neuronal circuitry underlying adaptive behavior for a whole class of vertebrates may seem, questions like this were being addressed by neuroanatomists a century ago. In fact, soon after Skinner was viewing brains as black boxes, Herrick [3] was summarizing the results of ablation studies in salamanders [115,116] similar to those performed in frogs by James [42]:

…in *Amblystoma* [sp.] between the early swimming and early feeding stages… the swimming movements, which in younger stages are perfectly coordinated by the [hindbrain-] spinal central apparatus alone, lose this autonomy, and participation of the midbrain is essential for the maintenance of efficient swimming… It is during this period that tecto-bulbar and tecto-spinal connections of essentially adult pattern are formed.

Today these studies are much simpler as audioradiograms [144], manganese-enhanced Magnetic Resonance Imaging (MEMRI [145]), and related techniques replace ablation as a tool for identifying CNS regions and nuclei responsible for interesting behaviors.

So, no, it is not necessary for behavioral ecologists to know the details of neural circuitry to understand amphibian responses to environmental change. But being scientists and therefore in the business of acquiring and disseminating knowledge, it is critical for behavioral ecologists to know that this circuitry exists, that much of it has been worked out, and that animals employ it to their benefit when making decisions [38,55,117].

### 7.4. Do Neuroanatomists Need to Know the Breadth of Amphibian Natural History?

The short answer is yes in that it provides perspective on the range of possibility [38,55]. While many neuroanatomical papers pay homage to natural history, few fully incorporate the breadth of available natural history knowledge. The idea of a scala natura, or the ladder-like progression from fish to frog to lizard to canary to monkey to human, remains strong in comparative studies of vertebrate nervous systems. But what does “the frog” mean when frogs are grouped into 57 families encompassing ~7750 species and each species has some feature—molecular, morphological, physiological, or behavioral—that makes it unique [6,146]? Anuran novelty includes aquatic, terrestrial, arboreal, and subterranean lifestyles; being tasty or toxic; having internal or external fertilization; locomotion that includes swimming, walking, hopping, and gliding; the ability to know a landscape well enough to exhibit philopatry; eggs that can be aquatic, terrestrial, or arboreal (e.g., in bromeliad tanks or foam nests); parental care including maternal provisioning, egg carrying, tadpole transport, or egg incubation (in skin pouches or the stomach); feeding as a generalist or specialist (often on ants or termites); coloration that can be camouflage, aposematic, or varied to match the background; calling synchronously or asynchronously; cocoon formation and water conservation using behavior (microsite selection) or behavioral physiology (waxy lipid secretions the frog spreads over its body [19,147]). Some of these variations in behavioral ecology are correlated with enhanced brain regions [146,148,149]. Exceptions to the scala natura approach in neuroanatomy include Mike Ryan and colleagues’ work on the Tungara Frog (*Engystomops pustulosus*) [150], Schlosser’s comparative work on the Common Coquí (*Eleutherodactylus coqui*) [151], Gerhard Roth’s work on a range of salamander and frog brains [34] (summarized in [152]), Wen Bo Liao’s research [146], and Sabrina Burmeister’s recent work [38,55].

The best example we know where a little natural history knowledge would have gone a long way comes from the auditory system of American Bullfrogs (*Rana catesbeiana*). Much is known about the auditory system in adult frogs—how it is built and how hormones can influence it (see above)—and much of it was discovered in Bullfrogs. For example, in an authoritative chapter in a book entitled *Neuroethological Studies of Cognitive and Perceptual Processes*, Simmons and Buxbaum address half of the book’s title by describing the perceptual processes of call production and processing in American Bullfrogs [153]. But, perhaps influenced by Schmidt [154], who reported that ablating the telencephalon and dorsal thalamus of American Toads (*Bufo americanus*) had no effect on eliciting male vocal responses or female phonotaxis (but see [100,155,156]), Simmons and Buxbaum never considered auditory processing beyond the midbrain torus semicircularis (forebrain diencephalic and telencephalic auditory circuitry exists and around the same time was being summarized [29,92]). Had Simmons and Buxbaum [153] contemplated the conditions under which American Bullfrogs call in nature, they might have continued their discussion rostrally into the forebrain and explored the second half of the book’s title—the cognitive processes underlying American Bullfrog advertisement calls—for the following reason.

As with many large-bodied ranids, American Bullfrogs use a resource-based mating system [83,157,158]. Males gather in a portion of a lake or pond and begin calling; females are then attracted to the area and choose their mate based on a male’s call characteristics and the relative quality of the territory he controls. Females are particularly attracted to the low frequency calls produced by large males. Therefore, in a breeding season that can last weeks or longer, large males get the majority of mating opportunities. Disadvantaged small males have non-calling ways to obtain mates including intercepting females as they approach large calling males [83,157]. These small males tend to be young first-time breeders and are called satellites [159] or parasites [83,157]. Several factors such as the size of dominant males, the number of males present, and the availability of receptive females determine whether a small, young male American Bullfrog ‘makes a decision’ to either call and defend a territory or become a satellite/parasite [83]. As Lucas and colleagues detail [160]:

“A decision can be thought of as a commitment of time to some particular course of action when an animal has more than one alternative behaviour available to it. The choice of behaviour should depend on the relative fitness payoffs accrued from the expression of each of the alternatives”. [161]

While American Bullfrog advertisement calls can be induced in the laboratory using evoked potential and playback calls [153], in nature, first year male Bullfrogs must decide whether or not to call based on their social situation [83,157,160]. We hypothesize this contingency game being played by young male American Bullfrogs is a cognitive function driven by telencephalic circuitry, and this circuitry primed by hormones drives a male’s decision to produce advertisement calls in the highly competitive arena of resource-based breeding.

### 7.5. What Does Acknowledging Amphibian Cognition Mean for the Understanding of Amphibian Behavior?

Appreciating the cognitive abilities of amphibians gives behavioral ecologists and conservation biologists deeper insights into the capabilities of their study animals and opens explanatory possibilities. For example, in the young male Bullfrog satellite/parasite scenario described above, female Bullfrogs are usually depicted as passive victims, but this is not the case. Rick Howard (pers. comm.) has observed when young males are absent, female Bullfrogs approach males on the surface of the water; in contrast, when young males are present, females dive, approach the large calling male underwater, and emerge next to him, thereby avoiding being mounted and wasting that year’s reproductive effort on a young male. Howard believes this is an example of cognition on the part of the females, and we agree.

As a component of our research on Crawfish Frogs, Jen Heemeyer used radiotransmitters to track frog movements during migrations to and from breeding wetlands [62,63]. She experimented with attaching transmitters to a frog’s waist using a belt, but for reasons of animal health (belts abrade sensitive belly skin) and evidence this apparatus was creating resistance (plant stems caught on the transmitter as frogs moved through dense prairie vegetation), she opted instead to implant transmitters into the abdominal cavity of her subjects. Recently, one of us reviewed a manuscript describing a study on a related species using belted transmitters where one conclusion from these authors was their animals had a higher than expected use of roadways. This ran contrary to much of the available literature on the negative relationship between amphibians and roads (i.e., roads are desiccating and expose amphibians to predators and traffic [162]). The authors of the manuscript being reviewed sought an ethological explanation for this atypical road use, which we will not detail here. However, had they understood that amphibians possess forms of spatial cognition [38,55,117], they would have considered the possibility that their telemetered frogs knew this object they were trailing (transmitter and antenna) was inhibiting them as they struggled through thick herbaceous vegetation (as Heemeyer found), and instead chose to approach their destination by traveling along roadways, which offered less resistance (if not more exposure).

### 7.6. What Does Acknowledging Amphibian Cognition Mean for Their Conservation?

Knowing that amphibians have the ability think their way around their landscape reinforces the importance of habitat in their conservation. Not only is it critical to have essential ecosystems conserved, there must be enough variation in core habitat types to accommodate climate extremes. For example, Crane Pond where Rick Howard made his Bullfrog observations had three lobes that differed in depth, and Bullfrog adults would select which lobe to use to breed based on water temperature. As Howard describes (pers. comm.) “early in the year, they went to the warmest lobe, late in the summer that lobe was too hot and they went to the coolest, deepest lobe. Mid-summer, they went to the lobe that was intermediate in temperature”.

Further, working in the Prairie Pothole Region of northwestern Iowa, we found that Northern Leopard Frogs (*Rana pipiens*) and Eastern Tiger Salamanders (*Ambystoma tigrinum*) shift their preference for breeding wetland type (seasonal, semipermanent, permanent) across drought cycles to ensure their larvae develop in fishless wetlands containing sufficient water to sustain them through to metamorphosis [163].

It is our job as behavioral ecologists interested in conserving the animals we study to work through the hows and whys of animal-animal and animal-environment interactions; as Herrick indicated [3], this is why we study brains. To get it right, we need to acknowledge amphibians have cognitive capabilities and in the context of the landscapes where they evolved can often find their own solutions to the challenges they face (if they could not, they would not have persisted). As Doug Chadwick [2] suggests, the more animals we get to know, the more we get to know animals.

## 8. Conclusions

Using data from our field research on Crawfish Frogs (15 years of data, field experiences, and peer-reviewed publications on this Indiana State Endangered and Globally Near-threatened species), we hypothesize (Example 1) their response to seasonal temperature and moisture variations, and the synchronicity of this response across solitary individuals can be explained through basic spinal cord-brainstem stimulus-response circuits; i.e., we find no proof of cognition. In contrast, the transitional calling behavior Crawfish Frogs exhibit (Example 2), their response to perceived threats (Example 3), and their variable, relative size-dependent responses to other organisms (Example 4) suggest Crawfish Frogs use cognition, or the mental process of acquiring knowledge and understanding through sensations, perceptions, and experiences, to respond to complex ecological and social situations. Given these results, we offer that progress in the field of neuroanatomy will be accelerated by expanding the range of study species to incorporate the vast life history and natural history diversity represented by amphibians, and that the field of behavioral ecology will be enriched by recognizing the adaptability of neuronal circuitry in basal vertebrates. It is especially important to gain as much of this big picture understanding as we can in species of conservation concern, in order to not lose the opportunity of ever being able to acquire this knowledge.

## Figures and Tables

**Figure 1 animals-15-00736-f001:**
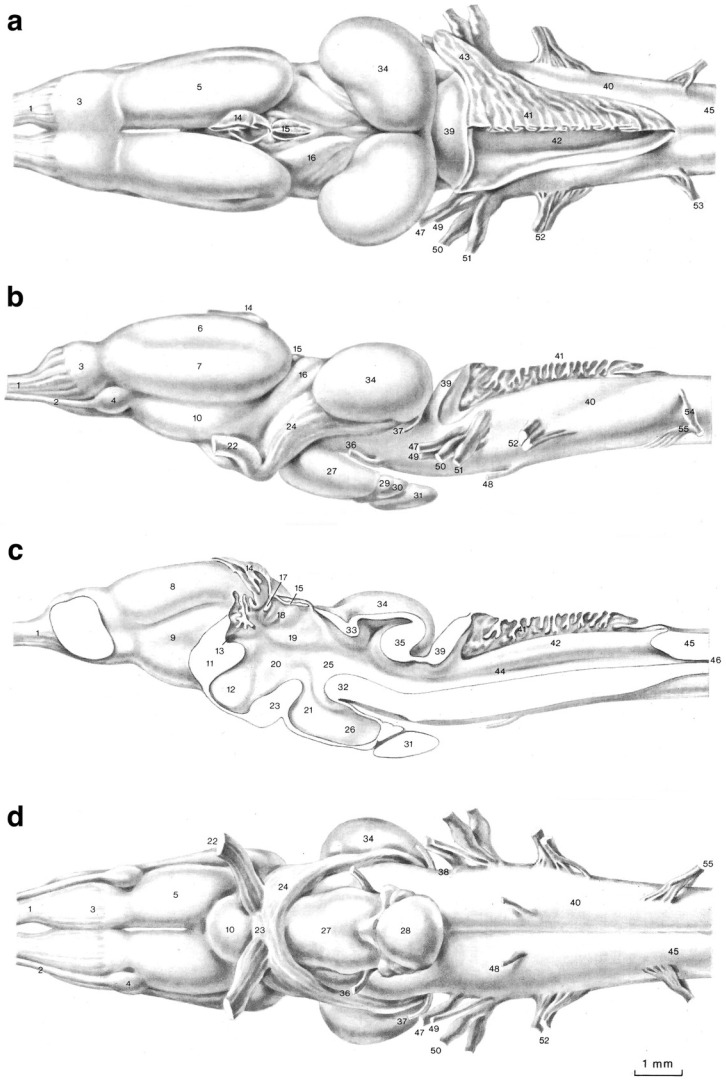
(**a**) Dorsal, (**b**) lateral, (**c**) midline, and (**d**) ventral views of the brain of an adult *Pelophylax esculentus*; rostral is to the left. For the purposes of this paper, we distinguish the telencephalon (numbers 5, 6, 7, 8, and 9) from the remainder of the brain. The key to structures: (1) Olfactory Nerve; (2) Vomeronasal Nerve; (3) Olfactory Bulb; (4) Accessory Olfactory Bulb; (5) Telencephalon; (6) Lateral Pallium; (7) Striatum; (8) Medial Pallium; (9) Septum; (10) Preoptic Area; (11) Anterior Commissure; (12) Preoptic Recess; (13) Interventricular Foramen; (14) Cerebral Paraphysis; (15) Cerebral Epiphysis; (16) Diencephalon; (17) Habenular Commissure; (18) Habenula; (19) Dorsal Thalamus; (20) Ventral Thalamus; (21) Hypothalamus; (22) Optic Nerve; (23) Optic Chiasm; (24) Optic Tract; (25) Third Ventricle; (26) Infundibular Recess; (27) Infundibulum; (28–31) Pituitary Gland; (32) Posterior Tuberculum; (33) Posterior Commissure; (34) Optic Tectum; (35) Torus Semicircularis; (36) Oculomotor Nerve (CN III); (37) Trochlear Nerve (CN IV); (38) Isthmic Fissure; (39) Cerebellum; (40) Rhombencephalon; (41) Choroid Plexus; (42) Fourth ventricle; (43) Lateral Recess of the Fourth Ventricle; (44) Sulcus Limitans; (45) Spinal Cord; (46) Central Canal; (47) Trigeminal Nerve (CN V); (48) Abducens Nerve (CN VI); (49) Facial Nerve (CN VII); (50,51) Vestibulocochlear Nerve (CN VIII); (52) Glossopharyngeal Nerve (CN IX); (53) Spinal Nerve (SN) 2; (54) Dorsal Root of SN 2; (55) Ventral Root of SN2. Modified from [29] and used with permission of Springer-Verlag, license number 5927050780492 issued 13 December 2024.

**Figure 2 animals-15-00736-f002:**
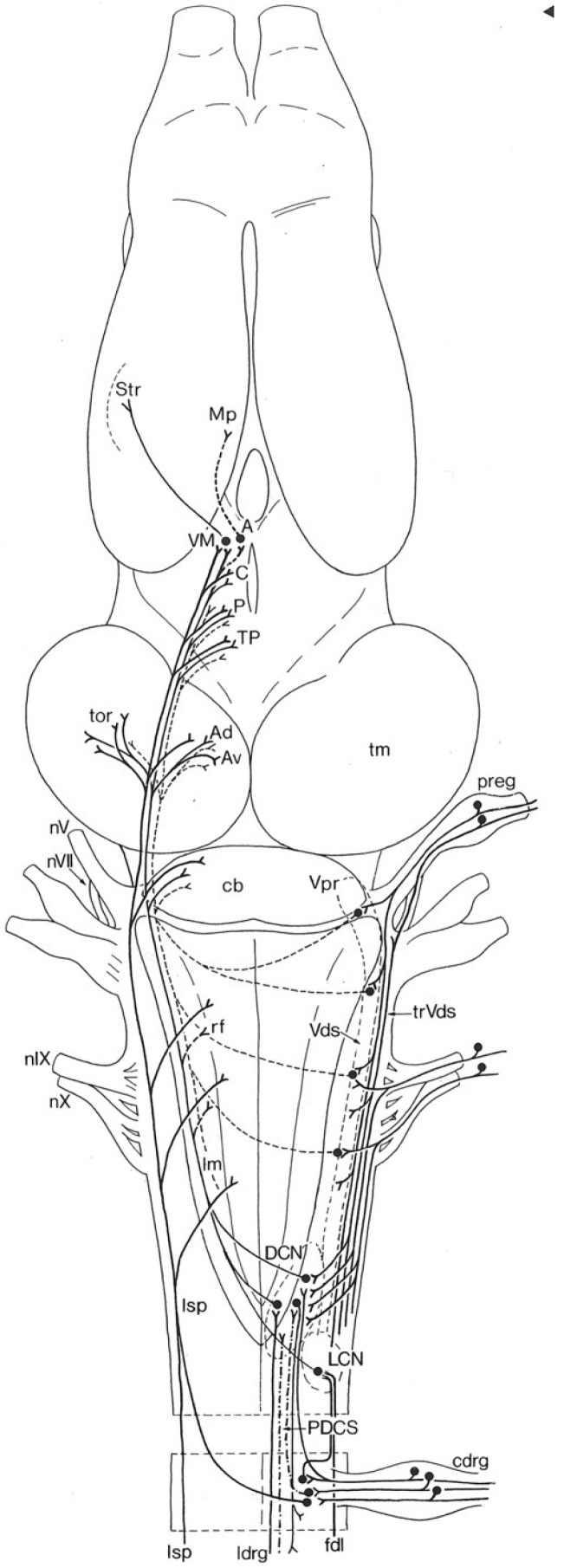
Schematic drawing of ascending somatosensory pathways from cutaneous receptors located in both the body and the head; rostral is up. Afferent nerves enter the spinal cord through the dorsal root ganglia (cdrg, lower right, only one spinal segment shown) and enter the brainstem through cranial nerves V, VII, IX and X. All primary afferents synapse ipsilaterally on second order neurons that decussate and terminate directly or indirectly in midbrain nuclei (torus semicircularis [tor] and tectum [tm]) and a subset of diencephalic thalamic nuclei (A, C, P, and TP). Third order neurons project from the anterior thalamic nucleus (A) to the telencephalic medial pallium (Mp), and from the ventromedial thalamic nucleus (VM) to the striatum (Str). The A → MP pathway is involved in memory formation, storage, and recall; the VM → Str pathway influences the onset of complex motor patterns. Image from [29] and used with permission of Springer-Verlag, license number 5927050780492 issued 13 December 2024.

**Figure 3 animals-15-00736-f003:**
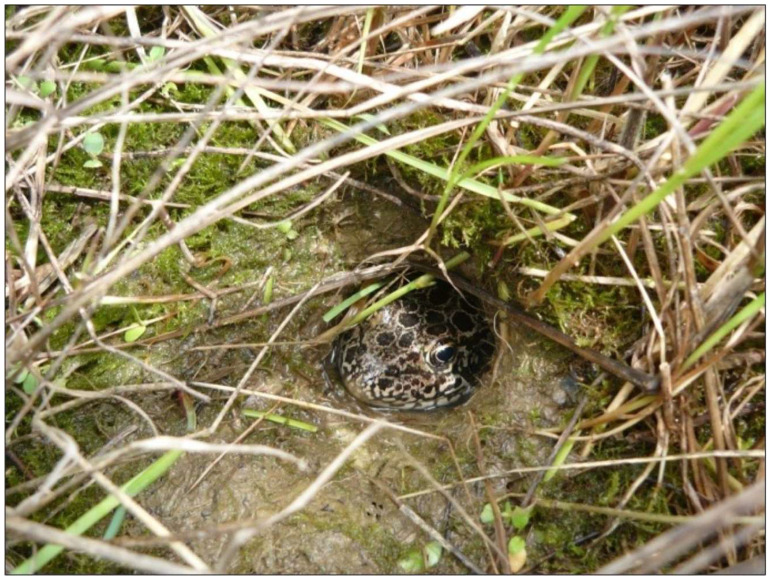
Adult Crawfish Frog (*Rana areolata*) in its primary crayfish-dug burrow. Note the snug fit of frog to burrow bore, and the tight congruence between the frog’s jawline and the curvature of the burrow. Photo taken by Nathan Engbrecht and used with permission.

**Figure 4 animals-15-00736-f004:**
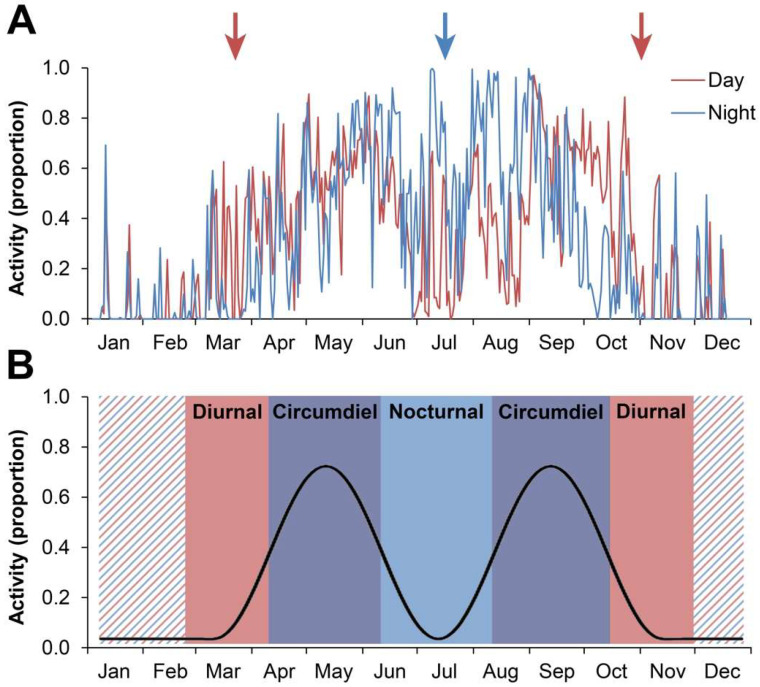
Crawfish Frogs (*Rana areolata*) shift activity patterns seasonally. (**A**) Mean percentage of time frogs (*n* = 17) spent at their burrow entrance or out on their feeding platform (i.e., active) during the day (red line) and at night (blue line) from 2009 to 2013; resolution is 5 min intervals captured around the clock by wildlife cameras positioned on each frog’s burrow. (**B**) Schematic representation of activity pattern shifts based on averaging and smoothing total activity (i.e., combined day and night data presented in (**A**) by month and colored to reflect when the most activity occurred (red, day; blue, night; purple, circumdiel; hashed, inactivity or breeding season migration). Some details and interpretations: (**A**) Red arrows and (**B**) Red shading: Early-spring increases (March–April) in activity and late fall decreases (October) in activity were due to frogs being primarily diurnal, likely due to cold nighttime temperatures. (**A**) Blue arrow and (**B**) Blue shading: During the hottest part of the summer (centered in July) frogs were primarily nocturnal, likely due to hot and dry daytime conditions. (**B**) Purple shading: Peak activity corresponded to seasons when environmental conditions allowed frogs to be active around-the-clock (circumdiel), which occurred in late spring (May) and early fall (September). For methodological details of this study see [66].

**Figure 5 animals-15-00736-f005:**
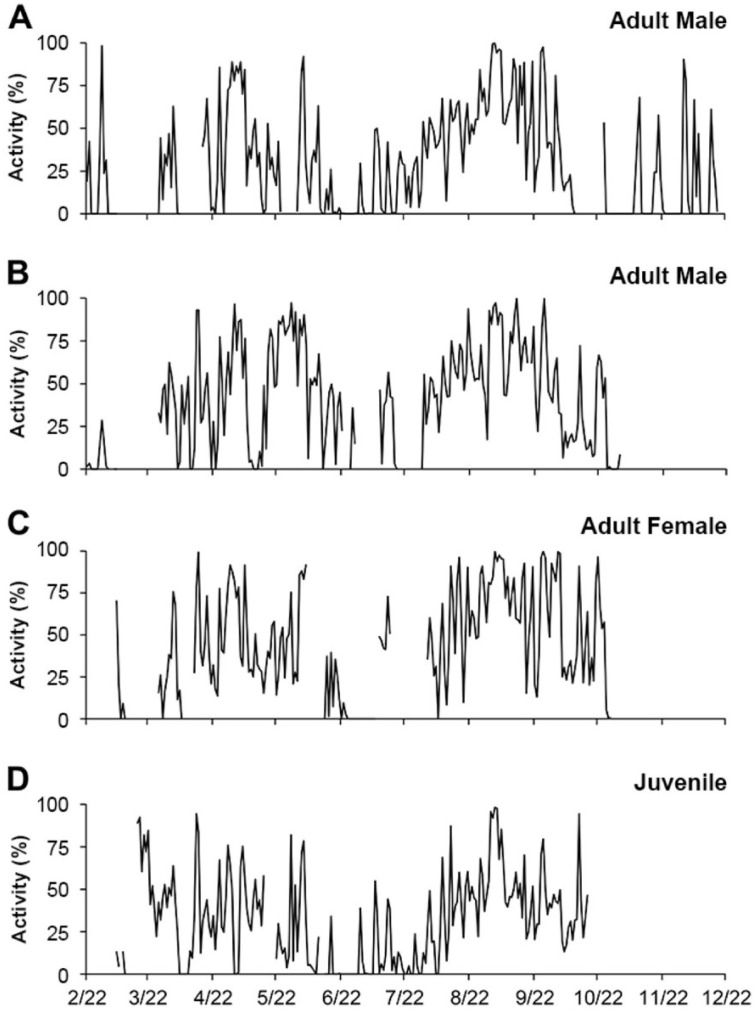
Daily activity of the four longest monitored Crawfish Frogs (*Rana areolata*) at Hillenbrand Fish and Wildlife Area-West (Greene County, IN, USA) in 2012. (**A**,**B**) adult males (Frogs 44 and 26, respectively); (**C**) an adult female (Frog 53); and (**D**) a juvenile. We measured activity as the percent of photos during which a frog was active (and photographed by a wildlife camera) on a given day. Notice the similarity in activity patterns across sexes and life stages. This figure first appeared in [66] and is reprinted with permission of Herpetological Conservation and Biology.

**Figure 6 animals-15-00736-f006:**
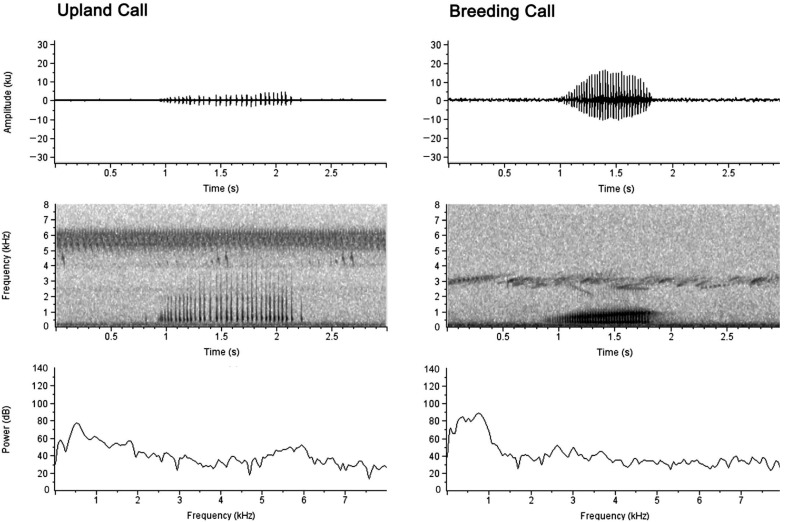
Waveforms, spectrograms, and spectrogram cross sections of a typical Crawfish Frog (*Rana areolata*) upland call (**left**) and a typical breeding call (**right**). Ambient temperatures for both calls were 16 °C. This figure is from [85] and is reproduced with permission of the American Society of Ichthyology and Herpetology.

**Figure 7 animals-15-00736-f007:**
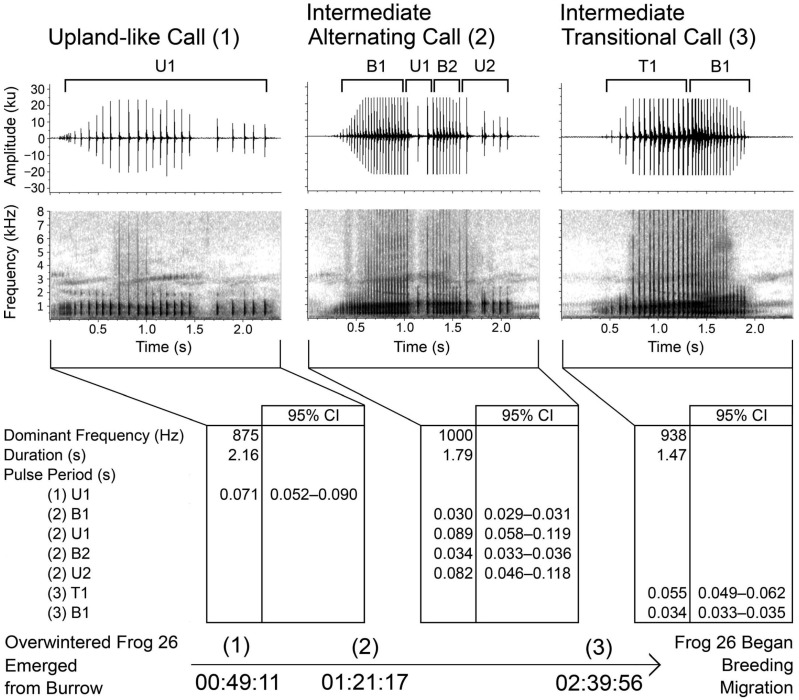
Frog 26’s (106 mm SVL) transition from a pre-breeding upland-like call to a pre-breeding call occurred over the course of two hours on 11 March 2013. This frog had just emerged from its burrow, emitted an upland-like call, then emitted various forms of intermediate calling before commencing its breeding migration. This figure is from [85] and is reproduced with permission of the American Society of Ichthyology and Herpetology.

**Figure 8 animals-15-00736-f008:**
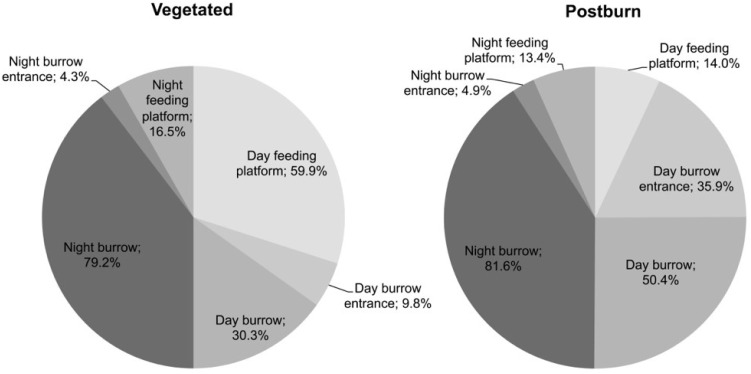
Percentage of time Crawfish Frogs (*Rana areolata*) spent on feeding platforms, at their burrow entrances, and in their burrows, day and night, in vegetated and postburn habitats. Note how nighttime percentages do not differ between vegetated and postburn areas, while, during the day, frogs in postburn areas spent more time at burrow entrances and in burrows. This Figure is reproduced from [101] and is included with permission of the American Association of Fire Ecology and Springer Nature.

**Figure 9 animals-15-00736-f009:**
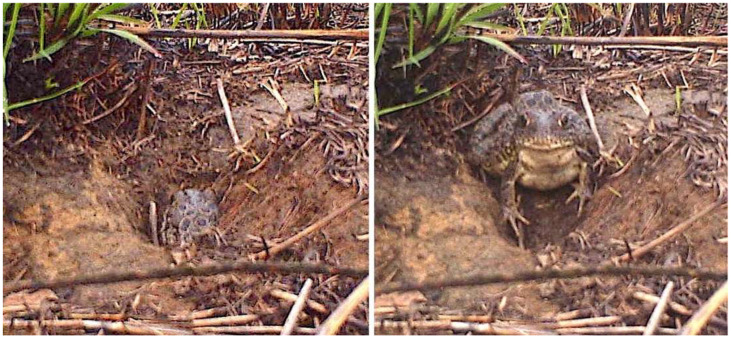
Wildlife camera images of Crawfish Frog 9 in a recently burned prairie positioned in the entrance to its burrow (**left**) and fully outside its burrow on its feeding platform (**right**) during the afternoon of 20 September 2011. The ratio of time a Crawfish Frog spends in the entrance to its burrow is five times higher in burned areas than vegetated areas, presumably to reduce its risk of predation by avian predators.

## Data Availability

Data may be accessed by contacting either author using the e-mail addresses listed above.

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
