# Peer review of "The Use of Cognition by Amphibians Confronting Environmental Change: Examples from the Behavioral Ecology of Crawfish Frogs (Rana areolata)"

_animals, 2025, doi:10.3390/ani15050736_

Round 1

Reviewer 1 Report

Comments and Suggestions for Authors

The core research question driving this study is to investigate the behavioral ecology of Crawfish Frogs, specifically focusing on how mental changes, such as stress or environmental challenges, affect their behavior including foraging, mating, and predator avoidanceThis manuscript offers detailed analysis and new perspectives on the behavioral adaptations of Crawfish Frogs by integrating  field observations, controlled laboratory experiments, and behavioral assays.

Specific Comments: 

Line 45: The main research question is clearly articulated, focusing on the challenge of understanding how mental changes affect the behavioral ecology of Crawfish Frogs. However, the introduction could benefit from a more detailed explanation of why this question is significant.

Line 125: The manuscript advances the current understanding by offering fresh perspectives and detailed analysis, surpassing existing studies in both breadth and depth. However, some comparisons with previous studies are lacking. 

Line 155: The methodology is generally robust, but refining data collection methods could enhance reliability. Additional controls, such as considering the influence of microhabitats, might be beneficial. 

Line 235: While the conclusions are largely consistent, some statements might overreach based on the data. Consider repositioning certain arguments to align more closely with the empirical findings.

Line 265: Consider to improve clarity in Figure 4 by providing a more detailed legend. Revise the legend for Figure 4 to include more specific information on the variables being plotted and any statistical significance. 

Overall, the study addresses a substantial gap in the literature and provides valuable insights into the behavioral ecology of Crawfish Frogs. Revisions that include a broader range of environmental stressors and reevaluate the conclusions to ensure they are supported by the data are necessary to strengthen the study.

Author Response

Reviewer #1

Yes

Can be improved

Must be improved

Not applicable

Does the introduction provide sufficient background and include all relevant references?

( )

(x)

( )

( )

Is the research design appropriate?

(x)

( )

( )

( )

Are the methods adequately described?

(x)

( )

( )

( )

Are the results clearly presented?

(x)

( )

( )

( )

Are the conclusions supported by the results?

( )

(x)

( )

( )

We believe we have addressed the Reviewer’s concerns regarding the Introduction and Conclusions—that these areas of the manuscript have now been improved, as follows.

The core research question driving this study is to investigate the behavioral ecology of Crawfish Frogs, specifically focusing on how mental changes, such as stress or environmental challenges, affect their behavior including foraging, mating, and predator avoidance. This manuscript offers detailed analysis and new perspectives on the behavioral adaptations of Crawfish Frogs by integrating field observations, controlled laboratory experiments, and behavioral assays. Thank you for the nice summary.

Specific Comments: 

Comment #1: Line 45: The main research question is clearly articulated, focusing on the challenge of understanding how mental changes affect the behavioral ecology of Crawfish Frogs. However, the introduction could benefit from a more detailed explanation of why this question is significant.

Response 1: We have modified this sentence to now read (modification in black), appears beginning on Line 40 in text: “One overlooked advantage amphibians possess in the struggle for survival, and one humans might use in their efforts to conserve them, is their brains share the same blueprint as human brains, which allows them to acquire knowledge and understanding through experiences—in other words amphibians have cognitive capabilities that assist them in their effort to survive.”

Comment 2: Line 125: The manuscript advances the current understanding by offering fresh perspectives and detailed analysis, surpassing existing studies in both breadth and depth.

Response#2: Thank you for the compliment. However, some comparisons with previous studies are lacking.  Without specific citations we do not know how to respond to this comment. We thought we were being thorough and with this in mind will note that our manuscript contains 28 pages of text and 17 pages of citations (~156 citations), which is about twice the number of citations expected for a manuscript of this length. Another reviewer has suggested adding specific citations, which we have done. If you would like to make suggestions, we would be happy to incorporate any papers we have missed.

Comment 3: Line 155: The methodology is generally robust, but refining data collection methods could enhance reliability. Additional controls, such as considering the influence of microhabitats, might be beneficial.

Response #3: The first three examples are from vetted, peer-reviewed, published work, and to be concise (this manuscript is already 63 pages long, including figures) we have left out details of these studies, assuming interested researchers will read these source papers. The final example is based on simple field observations and literature citations from other workers. If you or our editor feels these methods need to be elaborated, we will certainly do so.

Comment #4: Line 235: While the conclusions are largely consistent, some statements might overreach based on the data. Consider repositioning certain arguments to align more closely with the empirical findings.

Response #4: Another reviewer has made the same comment. We agree that we have overreached. We have now couched our interpretations as hypotheses everywhere in the manuscript—in each example and in the Discussion.

Comment #5: Line 265: Consider to improve clarity in Figure 4 by providing a more detailed legend. Revise the legend for Figure 4 to include more specific information on the variables being plotted and any statistical significance. We agree and thank the reviewer for their keen eye. We have re-written the legend, which now reads:

Figure 4. Crawfish Frogs (Rana areolata) shift activity patterns seasonally. A) Mean percentage of time frogs (n = 17) spent at their burrow entrance or out on their feeding platform (i.e., active) during the day (red line) and at night (blue line) from 2009–2013; resolution is 5-min intervals captured around the clock by wildlife cameras positioned on each frog’s burrow. B) Schematic representation of activity pattern shifts based on averaging and smoothing total activity (i.e., combined day and night data presented in A) by month and colored to reflect when the most activity occurred (red, day; blue, night; purple, circumdiel; hashed, inactivity or breeding season migration). Some details and interpretations: A) Red arrows and B) Red shading: Early-spring increases (March–April) in activity and late fall decreases (October) in activity were due to frogs being primarily diurnal, likely due to cold nighttime temperatures. A) Blue arrow and B) Blue shading: During the hottest part of the summer (centered in July) frogs were primarily nocturnal, likely due to hot and dry daytime conditions. B) Purple shading: Peak activity corresponded to seasons when environmental conditions allowed frogs to be active around-the-clock (circumdiel), which occurred in late spring (May) and early fall (September). For methodological details of this study see Stiles et al. (2017).

Comment #6: Overall, the study addresses a substantial gap in the literature and provides valuable insights into the behavioral ecology of Crawfish Frogs. Revisions that include a broader range of environmental stressors and reevaluate the conclusions to ensure they are supported by the data are necessary to strengthen the study.

Response #6: We thank the reviewer for this comment. Unfortunately, we are limited by our data, which here included responses to 1) climate variables; 2) site (upland vs. wetland) and conspecifics during breeding; 3) potential increased predation risk; and 4) to the same animal (snake) through ontogeny, but did not include responses to other environmental stressors. We hope that by showing behavioral ecologists their field data might shed light on amphibian cognition we soon might have other examples involving a diversity of species and their responses to a full range of environmental stressors.

Reviewer 2 Report

Comments and Suggestions for Authors

This paper presents an attempt to use examples based largely on detailed anecdotal observations (mostly with remote cameras) of the behavior of a species of frog in order to argue that behavior is often variable (within and between individuals), and largely driven by cognitive awareness of environmental conditions. The value of such an approach is convincingly made. However, the authors overstate the strength of their conclusions in a number of instances. The assumption that complex cognition is involved in control of the various behavioral variations observed is overstated in each case. They present hypotheses that the observed (or theorized) behaviors are being driven by perception-based awareness of the world around them. However, in each case a simpler hypothesis based on reflexive responses to variable environmental stimuli needs to be considered as well.

Overall the paper presents a useful summary of the idea that amphibians have complex brains and are at least potentially capable of integrating complex natural stimuli in order to develop cognitive awareness of their environment. The examples from crawfish frogs are less convincing than the authors suggest, but never-the-less provide a substrate for thoughtful consideration of how cognition in amphibians may play a critical role in their ability to respond to environmental variation. There is also some useful analysis of how the major brain pathways may contribute to this process.

I have made some comments on specific statement in the paper below.

Line 22-23.  This is an odd statement. Virtually every animal – vertebrate or invertebrate (consider bees for example) – ever studied has some capacity to learn and make behavioral adjustments. This line implies that this is special for amphibians. The capacity to learn and/or modify behavior in response to environmental cues is widespread and includes animals with vastly different neural architecture. Similarity to humans is not any sort of prerequisite for flexible behavior.

Line 52. What do the authors mean by “neatest?” Slang for “interesting?” I’m not sure non-native English speakers will know this slang.

Line 56-57. Probably the most devastating threat to amphibians are pathogens like the chytrid fungus, whose spread is like driven by climate change. Probably should be mentioned here

Line 65-66. This is a very odd statement. Amphibians share brain architecture with most vertebrates. Being similar to humans is neither special to amphibians nor a particular exceptional utility in survival. Among all groups of animals are many that species that exhibit great behavioral flexibility and others that show less flexibility. Amphibians are by no means special in this way. In other words, the similarities of brain architecture with humans is not special to amphibians, but simply a trait of the entire vertebrate lineage. I would simply leave out line 65.

Line 102-104. It would be helpful to distinguish between “cognition” as it is used in this paper and sensory perception. If a sensory stimulus (a call for example) triggers hormone release, and this modifies behavior, has “cognition” occurred? If not, why not? Specifically, what makes “cognition” unique and what observations are required for us to know it has occurred in an animal model?

Line 186 (and further on).  Normally the common name of a species is not capitalized unless it includes a proper noun. So, these should be named “crawfish frogs” rather than “Crawfish Frogs.”

Line 203. Do they switch burrows as they age and grow?

Figure 4. You should label and explain what is shown in 4A and 4B. I assume 4B is the result of some type of smoothing or temporal averaging but it is not stated in the Figure legend.

Lines 306-401. The argument that this somewhat complex and variable inter-twining of two call types is convincing evidence for cognition. The authors argue that this complex combination of calls could only arise from frogs being “aware” of the complex combination of the stimuli that normally produce each call type. There is another simpler hypothesis: each call type is driven by an environmental stimulus that triggers hormone release and acts on different brain areas. If stimuli exist for both call types simultaneously the behavioral output is likely to produce unpredictable combinations of the two call types. Thus, I would conclude that “cognition” might be involved, or there may be complexity arising from two different stimulus-reflex pathways in a time when both sensory stimulus conditions are present. It is not necessary to assume awareness of what these different stimuli represent. The authors need to make a more compelling argument if they offer this as strong evidence for a role of cognition. The fact that different individuals produced different outputs is similarly not convincing proof of awareness and choice. In nearly all species brain and physiological functions exhibit variation – likely due to genetic and/or experiential differences.

     The authors could certainly raise the question of whether these complex differences arise from cognitive awareness of stimuli or whether they simply represent complex interactions of reflex pathways with different thresholds in different individuals. They claim that they can clearly support one of these hypotheses over the other is weak.

Line 452-454. Why would the frogs be safer from visually oriented predators in the hotter part of the day? Explain.

Line 477.  Common Garter snakes should be written “common garter snakes.”

Line 494-495. In what sense is it “necessary” for the frog to “know” its own size. Frogs can judge distances to objects, and this means they could use visual angle of an object plus its distance to assess size. If the retina is designed to assess visual angle, a threshold visual angle (= too large) could be wired into the retina. As the frog grows spacing between the elements of the circuit could grow. This is speculative, but my point is that one cannot assume the frogs are aware of their body size in order for them to have the capacity to assess the threat from a snake.

I am not saying the proposed hypothesis is wrong, but it should be presented as a hypothesis, not a fact. There is no experimental proof offered to support the contention that the frogs respond differently to different sized snakes, and there is similarly no data indicating how they might make this distinction, so it remains a set of hypotheses not a demonstration of the principle.

Line 516-520. In each example cognition-drive behavior has been proposed as a possible explanation – a hypothesis. But in no case was it convincingly proved as the correct hypothesis. This paragraph overstates the strength of the conclusions, which are reasonable, but in no sense proven.

Author Response

Reviewer #2

Yes

Can be improved

Must be improved

Not applicable

Does the introduction provide sufficient background and include all relevant references?

(x)

( )

( )

( )

Is the research design appropriate?

( )

( )

( )

(x)

Are the methods adequately described?

(x)

( )

( )

( )

Are the results clearly presented?

( )

(x)

( )

( )

Are the conclusions supported by the results?

( )

( )

(x)

( )

We believe we have addressed the concerns of the reviewer concerning the presentation of our results, as follows. We will say in advance that working through these comments has been pleasant, like having a conversation over a beer following a seminar. Thank you.

Comment #1: This paper presents an attempt to use examples based largely on detailed anecdotal observations (mostly with remote cameras) of the behavior of a species of frog in order to argue that behavior is often variable (within and between individuals), and largely driven by cognitive awareness of environmental conditions. The value of such an approach is convincingly made. However, the authors overstate the strength of their conclusions in a number of instances. The assumption that complex cognition is involved in control of the various behavioral variations observed is overstated in each case. They present hypotheses that the observed (or theorized) behaviors are being driven by perception-based awareness of the world around them. However, in each case a simpler hypothesis based on reflexive responses to variable environmental stimuli needs to be considered as well.

Response #1: Having now had this pointed out to us, we absolutely agree. We have now couched our tentative conclusions as hypotheses in each example and in the Discussion.

Comment #2: Overall the paper presents a useful summary of the idea that amphibians have complex brains and are at least potentially capable of integrating complex natural stimuli in order to develop cognitive awareness of their environment. The examples from crawfish frogs are less convincing than the authors suggest, but never-the-less provide a substrate for thoughtful consideration of how cognition in amphibians may play a critical role in their ability to respond to environmental variation. There is also some useful analysis of how the major brain pathways may contribute to this process.

Response #2: Thank you for understanding our intent.

I have made some comments on specific statement in the paper below.

Comment #3: Line 22-23.  This is an odd statement. Virtually every animal – vertebrate or invertebrate (consider bees for example) – ever studied has some capacity to learn and make behavioral adjustments. This line implies that this is special for amphibians. The capacity to learn and/or modify behavior in response to environmental cues is widespread and includes animals with vastly different neural architecture. Similarity to humans is not any sort of prerequisite for flexible behavior.

Response #3: We have modified this sentence (Lines 67–69) to now read: “Further, we submit that acknowledging amphibians possess cognitive abilities can enrich interpretations of not only behavioral and ecological observations but also of neuroanatomical and neurophysiological results.”

An explanation: We made this statement in response to the deterministic approach used by neurophysiologists working in the 1970s and beyond. To them, and to quote Ewert (1976) “there are key stimuli that activate fixed patterns of behavioral responses—as a key is related to its lock (Uexküll, 1909; Lorenz, 1939; Marler and Hamilton, 1966; Tinbergen, 1972). Analysis and identification of key stimuli (releasers of instinctive action) may be accomplished on the basis of “preprogrammed” neuron circuiting. The key stimuli may be represented by means of response characteristics of so-called master units.  Supposedly, such fixed (or innate) recognition systems (IRMs) are located mainly in the subcortical region and play a major role among the lower vertebrates.

Comment #4: Line 52. What do the authors mean by “neatest?” Slang for “interesting?” I’m not sure non-native English speakers will know this slang.

Response #4: Ugh. We agree and have deleted.

Comment #5: Line 56-57. Probably the most devastating threat to amphibians are pathogens like the chytrid fungus, whose spread is like driven by climate change. Probably should be mentioned here. 

Response #5: We’ll meet you halfway, here. To address your concern, we’ve now listed a few causes of declines in the following sentence (Lines 82–83): “Some of the most evolutionarily novel amphibians on the planet have been driven to extinction (Tyler, 1983; Crump, 2000; http://www.amphibiaweb.org) by factors such as habitat loss, climate change, and disease (Mendelson et al., 2006; Green, 2020), each loss a tragedy.”

FYI, here are our papers on chytrid effects on Crawfish Frogs:

Kinney, V.C., J.L. Heemeyer, A.P. Pessier, and M.J. Lannoo. (2011) Seasonal pattern of Batrachochytrium dendrobatidis infection and mortality in Lithobates areolatus: Affirmation of Vredenburg’s “10,000 zoospore rule.” PLoS ONE 6(3): e1670 doi:10.1371/journal.pone.0016708.

Terrell, V.C.K., N.J. Engbrecht, A.P. Pessier, and M.J. Lannoo. (2014) Drought reduces chytrid fungus (Batrachochytridium dendrobatidis) infection intensity and mortality but not prevalence in adult Crawfish Frogs (Lithobates areolatus). Journal of Wildlife Diseases 50:56–62.

At our study site, chytrid is a cause of Crawfish Frog deaths, but not the cause for their decline (it’s habitat loss due to agricultural plowing and their fidelity to their burrow).

Moreover, the chytid question is controversial beyond belief (or maybe not, we know what it’s like to be a scientist and have ‘competitors’). Karen Lipps’ group says chytrid proceeds in a wave: it was not there, it shows up, frogs die. Others, including David Green, say chyrid is and has been everywhere, modern environmental conditions that stress frogs make this disease fulminant, that’s when it kills. We’ve not taken sides, we’ve got no dog in this fight.

Comment #6: Line 65-66. This is a very odd statement. Amphibians share brain architecture with most vertebrates. Being similar to humans is neither special to amphibians nor a particular exceptional utility in survival. Among all groups of animals are many that species that exhibit great behavioral flexibility and others that show less flexibility. Amphibians are by no means special in this way. In other words, the similarities of brain architecture with humans is not special to amphibians, but simply a trait of the entire vertebrate lineage. I would simply leave out line 65. Will this change work?

Response #6: We agree and have modified (Lines 96–97) as follows: “One overlooked advantage amphibians possess in the struggle for survival is that being vertebrates their brains share the same blueprint as human brains.”

Comment #7: Line 102-104. It would be helpful to distinguish between “cognition” as it is used in this paper and sensory perception. If a sensory stimulus (a call for example) triggers hormone release, and this modifies behavior, has “cognition” occurred? If not, why not? Specifically, what makes “cognition” unique and what observations are required for us to know it has occurred in an animal model?

Response #7: We define (lines 115–117) cognition following Shettleworth (2009) as “the mental process of acquiring knowledge and understanding through sensations, perceptions, and experiences (involving memory and learning.” Hopefully this is OK.

Comment #8: Line 186 (and further on).  Normally the common name of a species is not capitalized unless it includes a proper noun. So, these should be named “crawfish frogs” rather than “Crawfish Frogs.”

Response #8: Nomenclature is such a hornet’s nest. Ornithologists capitalize the common names of birds, and recently herpetologists have decided to do the same (see SSAR Circular No. 39: Moriarty, J.J. 2012. Scientific and Standard English Names …). The senior author went round and round with systematists about this back in 2005 and finally agreed to agree to whatever they decided, which is to capitalize common names. Ugh.

Comment #9: Line 203. Do they switch burrows as they age and grow?

Response #9: Yes, but we had 3 frogs occupy their burrows for at least 5 consecutive years. Part of switching burrows may be the requirement to fit the burrow (but they can to some degree create their own fit by enlarging their burrows when the ground is muddy simply by going in and out of them), part of this is undoubtedly that they foul them. The sides and bottoms of these burrows are lined with feces and insect parts. We’ve described long-occupied burrows as looking like the pit of a porta potty at a state fair. When this happens, frogs must ‘decide’ enough is enough and move on.

Comment #10: Figure 4. You should label and explain what is shown in 4A and 4B. I assume 4B is the result of some type of smoothing or temporal averaging but it is not stated in the Figure legend.

Response #10: Another Reviewer mentioned this also, we have re-written as follows.

Figure 4. Crawfish Frogs (Rana areolata) shift activity patterns seasonally. A) Mean percentage of time frogs (n = 17) spent at their burrow entrance or out on their feeding platform (i.e., active) during the day (red line) and at night (blue line) from 2009–2013; resolution is 5-min intervals captured around the clock by wildlife cameras positioned on each frog’s burrow. B) Schematic representation of activity pattern shifts based on averaging and smoothing total activity (i.e., combined day and night data presented in A) by month and colored to reflect when the most activity occurred (red, day; blue, night; purple, circumdiel; hashed, inactivity or breeding season migration). Some details and interpretations: A) Red arrows and B) Red shading: Early-spring increases (March–April) in activity and late fall decreases (October) in activity were due to frogs being primarily diurnal, likely due to cold nighttime temperatures. A) Blue arrow and B) Blue shading: During the hottest part of the summer (centered in July) frogs were primarily nocturnal, likely due to hot and dry daytime conditions. B) Purple shading: Peak activity corresponded to seasons when environmental conditions allowed frogs to be active around-the-clock (circumdiel), which occurred in late spring (May) and early fall (September). For methodological details of this study see Stiles et al. (2017).

Thanks for the careful eye.

Comment #11: Lines 306-401. The argument that this somewhat complex and variable inter-twining of two call types is convincing evidence for cognition. The authors argue that this complex combination of calls could only arise from frogs being “aware” of the complex combination of the stimuli that normally produce each call type. There is another simpler hypothesis: each call type is driven by an environmental stimulus that triggers hormone release and acts on different brain areas. If stimuli exist for both call types simultaneously the behavioral output is likely to produce unpredictable combinations of the two call types. Thus, I would conclude that “cognition” might be involved, or there may be complexity arising from two different stimulus-reflex pathways in a time when both sensory stimulus conditions are present. It is not necessary to assume awareness of what these different stimuli represent. The authors need to make a more compelling argument if they offer this as strong evidence for a role of cognition. The fact that different individuals produced different outputs is similarly not convincing proof of awareness and choice. In nearly all species brain and physiological functions exhibit variation – likely due to genetic and/or experiential differences.

     The authors could certainly raise the question of whether these complex differences arise from cognitive awareness of stimuli or whether they simply represent complex interactions of reflex pathways with different thresholds in different individuals. They claim that they can clearly support one of these hypotheses over the other is weak.

Thank you for this insight; we agree. We have modified our description to read as follows (Lines 344–372):

The anoetic hypothesis suggests each call type is driven by an environmental stimulus (location, conspecific calling) that releases hormones that act on different brainstem regions or nuclei. If stimuli exist for both call types simultaneously the behavioral output will produce unpredictable combinations of the two call types. The challenge to this hypothesis is its complexity is beyond what we understand brainstem circuitry acting alone can accomplish.

The noetic hypothesis is more aligned with James’s (1890) observations, as follows. Being at his burrow, Frog 26 included in each calling bout an appropriate upland-like call. But, primed by androgens and perhaps influenced by the proximity of nearby Crawfish Frog males producing breeding calls (Feng and Schul, 2007), he also included in each bout a breeding-like call. After completing these bouts, his drive to breed overcame his burrow fidelity and he began his breeding migration. In this interpretation, we hypothesize Frog 26’s transitional calling behavior was a product of his environment (being at his burrow), the drive to breed (seasonal hormone circulation), and social factors (hearing nearby calling males and perhaps promoting arousal and an endocrine response). Whether these transitional calls were produced by one pacemaker nucleus shifting states (the most likely scenario) or two different pacemakers competing (Barkan et al., 2018) is unknown, but this complex series of behaviors must be the product of at least three separate sensory-motor systems (vocal production, hearing conspecifics, and locomotion to the source of calling conspecifics) and cannot be explained by what we known about reflexive, basal, anoetic stimulus-response circuits. Instead, Frog 26 exhibited situational knowledge (the transition from upland to breeding calls), species knowledge (attention and attraction to other calling male Crawfish Frogs), and self-knowledge (the drive to migrate to the source of conspecific breeding calls suggests a ‘them and me’ understanding). Under this hypothesis, the integration of these behaviors represents a cognitive circuitry likely originating in the telencephalon. Why? Because its complexity is beyond what we understand brainstem circuitry alone can accomplish. Note that while this transition from an upland state to a breeding state was non-linear in its details, it was linear in function in that it began with a male Crawfish Frog producing upland-like calls at its burrow and ended in that male producing breeding calls entering a breeding wetland.

Comment #12: Line 452-454. Why would the frogs be safer from visually oriented predators in the hotter part of the day? Explain.

Response #12: In our interpretation, the increased activity of these frogs in the afternoon wasn’t due to avoiding predators as much as it was due to a hesitancy to emerge from their burrows each day, which shifted the center of their daily activity to the afternoon. We have re-worded this as a hypothesis, as follows (Line 416–422):

“Given this observation, we reject the anoetic idea that Crawfish Frogs in the burned area were responding in a deterministic way to temperature. Instead, based on the behavioral pattern we observed we hypothesize Crawfish Frogs in the burn were ‘reluctant’ to show themselves. In the morning when frogs in the vegetated areas were emerging from their burrows and positioning themselves on their feeding platforms, frogs in the burned area appeared hesitant to emerge, and when they did, they tended to expose only their head (Figure 9).”

Comment #13: Line 477.  Common Garter snakes should be written “common garter snakes.”

Response #13: Again, I’ve relied on the herpetologist’s Common Names recommendations. I don’t know why they combine to make the word “Gartersnake” but it’s that way with every snake species.

Comment #14: Line 494-495. In what sense is it “necessary” for the frog to “know” its own size. Frogs can judge distances to objects, and this means they could use visual angle of an object plus its distance to assess size. If the retina is designed to assess visual angle, a threshold visual angle (= too large) could be wired into the retina. As the frog grows spacing between the elements of the circuit could grow. This is speculative, but my point is that one cannot assume the frogs are aware of their body size in order for them to have the capacity to assess the threat from a snake.

Response #14: We understand the issue. Imagine you’re a Crawfish Frog with a medium-sized Gartersnake approaching ‘with intent.’ If you’re a small frog, and you know you’re a small frog, you dive into your burrow and ‘assume the position.’ If you’re a large frog, and you know you’re a large frog, you ignore the snake. This scenario is consistent with our field observations. We have, for example, wildlife camera images of medium-sized Gartersnakes eating small Crawfish Frogs, and large Crawfish Frogs and medium-sized Gartersnakes sharing a burrow.

Comment #15: I am not saying the proposed hypothesis is wrong, but it should be presented as a hypothesis, not a fact. There is no experimental proof offered to support the contention that the frogs respond differently to different sized snakes, and there is similarly no data indicating how they might make this distinction, so it remains a set of hypotheses not a demonstration of the principle.

Response #15: We have done this. This section now reads as follows (lines 466–484): “So how do adult Crawfish Frogs consider Gartersnakes? Critically, because a mistake could mean death, we hypothesize Crawfish Frogs must assess the Gartersnake’s size relative to their own size. For a large adult frog, small snakes represent potential prey, medium-sized snakes can safely be ignored, and large snakes are predators. But Crawfish Frogs live several years before they become a large adult (their maximum longevity is at least 10 years; Lannoo and Stiles, 2020a). To assess a snake based on its size, a Crawfish Frog must have some notion of its own size, and this notion must be continuously recalibrated as it grows. To a newly metamorphosed Crawfish Frog (~33 mm SVL, 4 gr) almost every snake no matter its size must be considered a predator. As the frog grows (adults in our studies averaged ~95 mm SVL and 95 gr), small- and medium-sized Gartersnakes can be ignored while large snakes continue to be threats. As mentioned above, Frog 401 was eaten by an 80 cm-long Gartersnake. For older, larger Crawfish Frogs (110 mm SVL, 110 gr), all medium- and large-sized Gartersnakes can stop being viewed as predators and small snakes may begin to be considered prey. Again, for these shifts in perspective to occur, we hypothesize Crawfish Frogs must have a sense of their absolute or relative size, and therefore possess an awareness of certain attributes contributing to their sense of self. Since we know of no deterministic hindbrain, midbrain, or diencephalic forebrain circuits that post-metamorphically switch states based on size or age, this knowledge must involve telencephalic circuits continuously uploading, storing, and downloading knowledge, which is cognition (Shettleworth, 2009).

Comment #16: Line 516-520. In each example cognition-drive behavior has been proposed as a possible explanation – a hypothesis. But in no case was it convincingly proved as the correct hypothesis. This paragraph overstates the strength of the conclusions, which are reasonable, but in no sense proven.

Response #16: Again, we agree and have softened each of our conclusions, both as described in our examples and summarized in the Discussion by couching them as hypotheses.

Reviewer 3 Report

Comments and Suggestions for Authors

Most amphibian species use the surface of land, the banks of water bodies, and water bodies themselves for orientation behavior. The species in this study uses crayfish burrows as shelters, which complicates their behavior. A description of the behavioral test for cognition or the parameters taken into account is also needed. For mammals, motor activity and cognitive functions are assessed using behavioral tests: "open field", "narrowing path", "elevated plus maze" and conditioned passive avoidance reactions. The habitat of Crawfish Frogs Rana areolata crayfish burrows are similar to artificial tests (mazes) and probably contributed to the evolutionary development of cognitive functions. This object Crawfish Frogs Rana areolata (Baird & Girard, 1852), as well as other amphibians with complex behavior, require comprehensive research.

Despite the topical issue, the article requires revision. 

 Comments:

Too many colorful quotes and slogan expressions instead of factual data. I think it is necessary to include material on the comparative study of behavior (Burmeister, 2022, doi: 10.1159/000522108) taking into account the physiological characteristics of amphibian behavior, metabolism, and the size of individuals, for example, the cane toad Rhinella marina (Linnaeus, 1758) that has existed since the Miocene. In most cases, the predator avoidance response dominates.

For example, about the decline of amphibians (lines 50-51). "Are amphibians declining? Yes, they most certainly are (Mendelson et al., 2006; Gallant et al., 2007; Bishop et al., 2012; Green et al., 2020; Luedtke, et al. 2023)."

Lines 173-177: “We are simply challenging any notion of amphibians as billiard balls being passively knocked around life’s felt surface by the cue stick of climate change. A concept that fails to acknowledge the benefits that amphibian telencephalic circuitry provide; advantages of this highly successful basal tetrapod group have employed in response to the challenges they have faced in the 3,650 or so millennia of their existence.”

A number of expressions and statements are not proven and argued, lines 465-467: “Instead, this behavior, based on self-awareness and perhaps memory (both a product of the telencephalon; Burmeister, 2022), constitutes a form of logic and can rightfully be considered cognitive.”

Factual inaccuracies:

Lines 78-79 of "Rana esculenta" according to the modern nomenclature of "Pelophylax esculentus (Linnaeus, 1758)"

For example, line 789 of "13. Burmeister, S.S. 2022. Brain-behavior relationships of cognition in vertebrates: lessons from amphibians. Chapter 3. Advances in the Study of Behavior 54:109–127."

Line 803. The "DOI" is indicated:

"22. Damasio, A.R., and H. Damasio. 2022. Homeostatic feelings and the biology of consciousness. Brain 145:2231–2235. https://doi.org/10.1093/brain/awac194."

Link:

Burmeister SS. Ecology, Cognition, and the Hippocampus: A Tale of Two Frogs. Brain Behav Evol. 2022;97(3-4):211-224. doi: 10.1159/000522108. Epub 2022 Jan 20. PMID: 35051940.

Author Response

Reviewer #3

Yes

Can be improved

Must be improved

Not applicable

Does the introduction provide sufficient background and include all relevant references?

( )

(x)

( )

( )

Is the research design appropriate?

( )

( )

(x)

( )

Are the methods adequately described?

( )

( )

(x)

( )

Are the results clearly presented?

( )

( )

(x)

( )

Are the conclusions supported by the results?

Yes

Can be improved

Must be improved

Not applicable

Does the introduction provide sufficient background and include all relevant references?

( )

(x)

( )

( )

Is the research design appropriate?

( )

( )

(x)

( )

Are the methods adequately described?

( )

( )

(x)

( )

Are the results clearly presented?

( )

( )

(x)

( )

Are the conclusions supported by the results?

We believe we have addressed to concerns raised by this Reviewer. We note that some of these are philosophical and may not be resolvable.

Comment #1: Most amphibian species use the surface of land, the banks of water bodies, and water bodies themselves for orientation behavior. The species in this study uses crayfish burrows as shelters, which complicates their behavior. A description of the behavioral test for cognition or the parameters taken into account is also needed. For mammals, motor activity and cognitive functions are assessed using behavioral tests: "open field", "narrowing path", "elevated plus maze" and conditioned passive avoidance reactions. The habitat of Crawfish Frogs Rana areolata crayfish burrows are similar to artificial tests (mazes) and probably contributed to the evolutionary development of cognitive functions. This object Crawfish Frogs Rana areolata (Baird & Girard, 1852), as well as other amphibians with complex behavior, require comprehensive research.

Response #1: Thank you for this perspective. We believe students of the history of biology will agree that model systems (with an emphasis on depth of knowledge) have given great insights into the workings of organisms. But those who study diversity (i.e., breadth of knowledge) have also given us great insights (i.e., Charles Darwin). The modern tendency is for these two approaches to be in conflict (i.e., depth is better than breadth or vice versa), but in fact the approach used in modeling systems and the approach used by recognizing diversity within and across taxa complement each other and provide a fuller picture of life (both breadth and depth) than either approach in isolation.  We explain ourselves in Lines 169–175: “Our goal here, suggestive of Burmeister’s (2022a, b) approach, is to follow Edelman et al.’s (2005) plea to use alternate strategies for assessing consciousness by expanding the catalogue of examples of amphibian cognition into the natural world of ecology—amphibians in the wild interacting with their biotic and abiotic environment (i.e., cognitive ecology; Mettke-Hofmann, 2014; Vonk, 2016; Moss and Shettleworth, 2019). Specifically, we ask in what situations do amphibians use cognition to respond to ecological and social challenges?”

Big picture: If one wanted to understand cognition in amphibians, the current model systems approach, based, as you point out, primarily on frogs with complex life histories give you a deep understanding of frogs with this lifestyle but very little understanding of what frogs that are fully aquatic, fully terrestrial, fully arboreal, or primarily subterranean do. If we truly want to understand what cognition in frogs means, we must study both the depth and breadth of what it means to be a frog. As another reviewer has stated about our ms.: The examples from Crawfish Frogs … provide a substrate for thoughtful consideration of how cognition in amphibians may play a critical role in their ability to respond to environmental variation. There is also some useful analysis of how the major brain pathways may contribute to this process.

Despite the topical issue, the article requires revision. 

 Comments:

Comment #2: Too many colorful quotes and slogan expressions instead of factual data. I think it is necessary to include material on the comparative study of behavior (Burmeister, 2022, doi: 10.1159/000522108) taking into account the physiological characteristics of amphibian behavior, metabolism, and the size of individuals, for example, the cane toad Rhinella marina (Linnaeus, 1758) that has existed since the Miocene. In most cases, the predator avoidance response dominates.

Response #2: Our colorfulness is intentional. Anyone with access to the news and a sense of history understands that the United States (at minimum) is descending into a new dark ages. A time when superstition and belief systems take precedence over facts and their application by the scientific method. Our future federal Health and Human Services Director believes injecting bleach is more effective than vaccines against disease. Why is this happening? In part because the STEM disciplines are viewed as dull, boring, and practiced by people who don’t otherwise ‘have a life.’ Whereas fundamentalist megachurches are seen as active, vibrant, and ‘cool.’ “Want to meet interesting and exciting people? Go to a megachurch not a chemistry lab.” Science is currently losing the culture war.

A tragedy is on the horizon. Polio and other diseases now held in check by vaccines will be making a comeback. The most likely way this trend will be reversed will be when people, especially children, begin to die as a result. In the meantime, it doesn’t hurt to share with others the joy we have in doing science. The people we cite are world class scientists (Gould, Herrick) or science communicators (Chadwick). Other scientists have also been colorful (Oppenheimer, Feynman, David Suzuki, Hawking, Asimov, Nye, Attenborough) which to our knowledge has not diminished their reputations as conveyors of objective truth. While we are not the scientists or communicators these people have been, we follow their lead in trying to make our science dynamic and interesting, and we do not believe our being colorful has interfered with us being objective.

The three quotes we use hit on the major themes that run through the manuscript.

To our knowledge the only ‘slogan’ we’ve used is ‘homologous nuclei have homologous functions.’ Which to our minds is less a slogan than it is an evolutionary principle. The senior author learned it from Ted Bullock and Glenn Northcutt while postdoc-ing in Walter Heiligenberg’s lab in the late 1980s. This principle is also the rationale behind the thinking in Streidter and Northcutt’s 2020 book on vertebrate brain evolution.

We have deleted the word “neatest.”

We are huge fans of Burmeister and in fact cite her 2022 chapter several times here. What we admire about her work is that she broke free from the constraints of behavioral ecology to address neuroanatomy, and broke free from comparative neuroanatomy to address field-based behavioral ecology. She behaves like she is working in 1948, before disciplines split off and became insular. This is our aim, also.

We compiled the literature on Crawfish Frogs in our 2020 book. Before our work, Hobart Smith described Crawfish Frogs as the “Most secretive ranid in North America.” This reticent species has never lent itself to physiological studies, so we cannot incorporate this type of information here. Plus, where we work, in Indiana, Crawfish Frogs are a state-endangered species, with the consequence that no invasive experiments can be done on them. Here, we do consider Crawfish Frog behavior (uniquely, since are one of the few species of frogs that can be followed behaving naturally in the wild using wildlife cameras). We also consider animal size in Example 4.

Comment #3: For example, about the decline of amphibians (lines 50-51). "Are amphibians declining? Yes, they most certainly are (Mendelson et al., 2006; Gallant et al., 2007; Bishop et al., 2012; Green et al., 2020; Luedtke, et al. 2023)."

Response #3: This is true. The senior author here is co-author on three of these papers. The debate around this issue can be surprisingly contentious.

Comment #4: Lines 173-177: “We are simply challenging any notion of amphibians as billiard balls being passively knocked around life’s felt surface by the cue stick of climate change. A concept that fails to acknowledge the benefits that amphibian telencephalic circuitry provide; advantages of this highly successful basal tetrapod group have employed in response to the challenges they have faced in the 3,650 or so millennia of their existence.”

Response #4: This is also true. In places in the Introduction we have intentionally followed the somewhat causal style of scientific writing employed by Platt (1964; Strong Inference), and Damasio (2001, Fundamental Feelings) in their Science papers in order to make our paper more accessible to non-specialists. See also our comments, above.

Comment #5: A number of expressions and statements are not proven and argued, lines 465-467: “Instead, this behavior, based on self-awareness and perhaps memory (both a product of the telencephalon; Burmeister, 2022), constitutes a form of logic and can rightfully be considered cognitive.”

Response #5: We agree. The other reviewers have made the same comment, and we have softened our conclusions both in the examples and in the Discussion by couching them as hypotheses. Thank you for pointing this out.

Factual inaccuracies:

Comment #6: Lines 78-79 of "Rana esculenta" according to the modern nomenclature of "Pelophylax esculentus (Linnaeus, 1758)"

Response #6: We have noted and made (Line 1150) this nomenclatural change. Thank you.

Comment #7: For example, line 789 of "13. Burmeister, S.S. 2022. Brain-behavior relationships of cognition in vertebrates: lessons from amphibians. Chapter 3. Advances in the Study of Behavior 54:109–127."

Response #7: We have checked this reference and added the doi (Line 788). Thank you.

Comment #8: Line 803. The "DOI" is indicated:

"22. Damasio, A.R., and H. Damasio. 2022. Homeostatic feelings and the biology of consciousness. Brain 145:2231–2235. https://doi.org/10.1093/brain/awac194."

Response #8: We have checked this reference (Lines 806–807). Thank you.

Comment #9: Link:

Burmeister SS. Ecology, Cognition, and the Hippocampus: A Tale of Two Frogs. Brain Behav Evol. 2022;97(3-4):211-224. doi: 10.1159/000522108. Epub 2022 Jan 20. PMID: 35051940.

Response #9" Thank you for this reference. We have incorporated it into the manuscript in several places (Lines 169, 441, 557, 616, 620, 641, 711).

Round 2

Reviewer 2 Report

Comments and Suggestions for Authors

The authors have addressed my concerns adequately. I do not recommend any further changes.

Author Response

Reviewer #2

Yes

Can be improved

Must be improved

Not applicable

Does the introduction provide sufficient background and include all relevant references?

(x)

( )

( )

( )

Is the research design appropriate?

( )

( )

( )

(x)

Are the methods adequately described?

(x)

( )

( )

( )

Are the results clearly presented?

(x)

( )

( )

( )

Are the conclusions supported by the results?

(x)

( )

( )

( )

Comments and Suggestions for Authors

The authors have addressed my concerns adequately. I do not recommend any further changes. Thank you once again for your constructive comments.

Reviewer 3 Report

Comments and Suggestions for Authors

The authors have made changes and improved the publication.

Exception: Comment #2: Too many colorful quotes and slogan expressions instead of factual data. I think it is necessary to include material on the comparative study of behavior (Burmeister, 2022, doi: 10.1159/000522108) taking into account the physiological characteristics of amphibian behavior, metabolism, and the size of individuals, for example, the cane toad Rhinella marina (Linnaeus, 1758) that has existed since the Miocene. In most cases, the predator avoidance response dominates.

It seems to me that to analyze the cognitive abilities of amphibians, a review of data and facts is necessary.

Thus, views on the cognitive abilities of amphibians changed in accordance with the change of scientific paradigms, for example, Thomas Kuhn's "The Structure of Scientific Revolutions" (Kuhn, 1962).

At the stage of dominance of the "conditioned reflex" paradigm, it was concluded that frogs have no figurative behavior and typical conditioned reflex activity. At the same time, amphibians are higher than fish on the phylogenetic ladder, and the functional development of their brain is much lower than fish. This conclusion agrees well with the well-known histological study by Herrick (1927), who found that the amphibian brain has undergone reverse development and, as a result, their brain is less differentiated. These conclusions were confirmed by the experiments: "During the croaking that was evoked in another frog, the experimental frog was electrically stimulated. A conditioned reflex to croaking was formed in this frog, but it was developed very slowly. After many dozens of combinations, having heard the croaking, the experimental frog moved to another section. But this reflex did not strengthen, did not become constant even after 500-600 combinations. The same protective reflex was not developed at all for other sounds (Beritashvili, 1929). Karamyan (1966), who studied the formation of associative temporary connections between simultaneous or successive indifferent stimuli, such as sound and light, in amphibians (toads, axolotls, etc.), came to a similar conclusion regarding the mechanisms of conditioned reflex activity in amphibians (toads, axolotls, etc.). In amphibians, such stimuli did not form temporary connections even after hundreds of combinations.

Later, the emphasis was placed on the complexity of walking control and orientation behavior, including in the study of amphibians. It seems to me that the authors should consider the change in ideas in the aspect of a paradigm shift. The same Herrick CJ. The Brain of the Tiger Salamander (Herrick, 1948) The University Of Chicago Press, Chicago, Illinois, notes “Now that this isolationism has given way to genuine collaboration among specialists in all related fields - physiology, biochemistry, biophysics, clinical practice, neuropathology, psychology, among others - we witness today a renaissance of the science of neurology. The results of the exacting analytical investigations of the specialists can now be synthesized and given meaning. The task of comparative anatomy in this integrated program of research is fundamental and essential. " « Quadrupedal locomotion is a very complicated activity compared with the simple mass movement of swimming. The action of the four appendages and of every segment of each of them must be harmoniously coordinated, with accurate timing of the contraction of many small muscles. These local activities are "partial patterns" of behavior, in Coghill's sense. From the physiological standpoint there is great advance, in that the primitive total pattern is supplemented, and in higher animals largely replaced, by a complicated system of coordinated partial patterns. This is emphasized here because it provides the key to an understanding of many of the differences between the nervous systems of fishes, salamanders, and mammals. At the same time, in his scientific works he does not discuss philosophical issues, but speaks about the development of ideas and the complexity of research, for example: "The experimentalist must know exactly what he has done to the living tissue before he can interpret his experiment. In the past it too often happened that a physiologist would stab into a living frog, take his kymograph records, and then throw the carcass into the waste-jar. This is no longer regarded as good physiology. Without the guidance of accurate anatomical knowledge, sound physiology is impossible; and, without skillful physiological experimentation, the anatomical facts are just facts and nothing more.»

People tend to perceive information that corresponds to their formed paradigm - beliefs, and ignore the opposite. This is confirmation bias: it leads to polarization, distorted perception of reality and erroneous decisions. Confirmation bias is one of the most common cognitive distortions.

In science, facts are a realized experiment, experience and repeatability (experiment) criterion.

P.S.

" It is interesting that the same author (Herrick ) wrote a book: "Fatalism or Freedom: A Biologist's Answer" (Herrick, 1926): "This excursion by a biologist into a field usually posted "No Trespassing" from this side has a two-fold purpose: first, to inquire how far it is profitable to push the inquiry into the problems of human behaviour with the ordinary methods of natural science and so to knit our conscious and social life in with the rest of our world of experience in lawful fashion; and second, to see whether the conclusions thus reached shed any light upon the most acute problems of human conduct selfcontrol, self-determination, self-culture, social control, personal and social morality."

Author Response

Reviewer #3

Yes

Can be improved

Must be improved

Not applicable

Does the introduction provide sufficient background and include all relevant references?

( )

(x)

( )

( )

Is the research design appropriate?

( )

(x)

( )

( )

Are the methods adequately described?

( )

(x)

( )

( )

Are the results clearly presented?

( )

( )

(x)

( )

Are the conclusions supported by the results?

( )

(x)

( )

( )

Thank you once again for your review.

Big picture, we now understand the difference in our perspectives. The scientific process as generally practiced proceeds from observation (field or lab) to hypothesis, to hypothesis testing (including experimentation), to theory. We understand that you often work at the back end of this process (hypothesis testing through experimentation, and theory formation) while we work at the front end (observations and hypothesis formation). History tells us that both ends of this spectrum by themselves are legitimate scientific pursuits. Many scientists (Darwin, Einstein) have worked at the front end of this process.

Comments and Suggestions for Authors

The authors have made changes and improved the publication.

Exception: Comment #2: Too many colorful quotes and slogan expressions instead of factual data. I think it is necessary to include material on the comparative study of behavior (Burmeister, 2022, doi: 10.1159/000522108) taking into account the physiological characteristics of amphibian behavior, metabolism, and the size of individuals … We agree about the importance of Burmeister’s work and have now featured Burmeister (2022b) to a greater degree. We cite her work 19 times.

We have incorporated all known relevant Crawfish Frog behaviors in our 4 examples. Again, this reticent species has never lent itself to physiological studies, so we cannot incorporate this type of information here. Plus, where we work, in Indiana, Crawfish Frogs are a state-endangered species, with the consequence that no laboratory observations or invasive experiments can be done on them. Here, we do consider Crawfish Frog behavior in the wild (uniquely, since are one of the few species of frogs that can be followed behaving naturally in the wild using wildlife cameras). We also consider animal size in Example 4.

for example, the cane toad Rhinella marina (Linnaeus, 1758) that has existed since the Miocene. In most cases, the predator avoidance response dominates. Yes, but Cane Toads are so much better known than Crawfish Frogs. As we state in the paper (Lines 641–644) “But what does “the frog” mean when frogs are grouped into 57 families encompassing ~7,750 species and each species has some feature—molecular, morphological, physiological, or behavioral—that makes it unique (Liao et al., 2015; http://www.amphibiaweb.com).” Burmeister’s results reinforce this idea. Further, another reviewer has stated about our ms.: The examples from Crawfish Frogs … provide a substrate for thoughtful consideration of how cognition in amphibians may play a critical role in their ability to respond to environmental variation. There is also some useful analysis of how the major brain pathways may contribute to this process.

In addition, we must consider that Crawfish Frogs are a near-threatened species. One of its sister species would today be extinct if not for the heroic efforts of two Zoos. The other sister species has been petitioned for federal listing. We owe it to posterity to document as much as we can about the biology of species we may lose forever.

It seems to me that to analyze the cognitive abilities of amphibians, a review of data and facts is necessary. We have addressed this issue by adding two sections. First, by putting (small scale) the brains of amphibians into an evolutionary context (Lines 96–105), second by including a most robust description of Burmeister’s (2022b) work (Lines 179–184). Beyond this, we do not know that we have missed citing any references that directly consider amphibian cognition.

Thus, views on the cognitive abilities of amphibians changed in accordance with the change of scientific paradigms, for example, Thomas Kuhn's "The Structure of Scientific Revolutions" (Kuhn, 1962). We simply do not have the space to include a detailed history of scientific thought in our paper. We, for example, follow John Platt’s notion of Strong Inference (1964, Science 146:347–353) when employing inductive reasoning, and when experimenting follow Karl Popper’s notion of hypothesis rejection, but with this work we are only at Platt’s stage of forming hypotheses. We consider this approach in more detail, below.

At the stage of dominance of the "conditioned reflex" paradigm, it was concluded that frogs have no figurative behavior and typical conditioned reflex activity. At the same time, amphibians are higher than fish on the phylogenetic ladder, and the functional development of their brain is much lower than fish. This conclusion agrees well with the well-known histological study by Herrick (1927), who found that the amphibian brain has undergone reverse development and, as a result, their brain is less differentiated. Thank you for this observation. Yes, the brains of modern amphibians, especially salamanders and caecilians, are neotenic, we have now addressed this in Lines 96–105.

These conclusions were confirmed by the experiments: "During the croaking that was evoked in another frog, the experimental frog was electrically stimulated. A conditioned reflex to croaking was formed in this frog, but it was developed very slowly. After many dozens of combinations, having heard the croaking, the experimental frog moved to another section. But this reflex did not strengthen, did not become constant even after 500-600 combinations. This situation may be different in nature, see Bee, M.A., V.T. Marshall, S.C. Humfield, and H.C. Gerhardt. 2002. The role of learning in the formation of frog choruses. Integrative and Comparative Biology 42:1193. The same protective reflex was not developed at all for other sounds (Beritashvili, 1929). Karamyan (1966), who studied the formation of associative temporary connections between simultaneous or successive indifferent stimuli, such as sound and light, in amphibians (toads, axolotls, etc.), came to a similar conclusion regarding the mechanisms of conditioned reflex activity in amphibians (toads, axolotls, etc.). In amphibians, such stimuli did not form temporary connections even after hundreds of combinations. Again, this situation may be different in the wild. For example for Crawfish Frogs see Engbrecht, N.J., J.L. Heemeyer, C.G. Murphy, R.M. Stiles, J.W. Swan, and M.J. Lannoo. (2015) Upland calling behavior in Crawfish Frogs (Lithobates areolatus) and calling triggers caused by noise pollution. Copeia 103:1046–1057.

Later, the emphasis was placed on the complexity of walking control and orientation behavior, including in the study of amphibians. It seems to me that the authors should consider the change in ideas in the aspect of a paradigm shift. The same Herrick CJ. The Brain of the Tiger Salamander (Herrick, 1948) The University Of Chicago Press, Chicago, Illinois, notes “Now that this isolationism has given way to genuine collaboration among specialists in all related fields - physiology, biochemistry, biophysics, clinical practice, neuropathology, psychology, among others - we witness today a renaissance of the science of neurology. The fields of comparative anatomy and behavioral ecology have since diverged and specialized. Unlike in Herrick’s time, it is rare today to see neuroanatomy tied to behavior (e.g., Strieder and Northcutt, 2020) and it is rare to see neuronal substrates cited in behavioral ecology papers. As we point out (Lines 655–659) valuable exceptions include Mike Ryan and colleagues’ work on the Tungara Frog (Engystomops pustulosus; Ryan, 1985), Schlosser’s (2008) comparative work on the Common Coquí (Eleutherodactylus coqui), Gerhard Roth’s work on a range of salamander and frog brains (summarized in Roth and Wake, 2001; see also Roth, 2013; Liao et al., 2015), and Burmeister’s work (Burmeister, 2022a, b).

The results of the exacting analytical investigations of the specialists can now be synthesized and given meaning. The task of comparative anatomy in this integrated program of research is fundamental and essential. " « Quadrupedal locomotion is a very complicated activity compared with the simple mass movement of swimming. The action of the four appendages and of every segment of each of them must be harmoniously coordinated, with accurate timing of the contraction of many small muscles. These local activities are "partial patterns" of behavior, in Coghill's sense. From the physiological standpoint there is great advance, in that the primitive total pattern is supplemented, and in higher animals largely replaced, by a complicated system of coordinated partial patterns. This is emphasized here because it provides the key to an understanding of many of the differences between the nervous systems of fishes, salamanders, and mammals. At the same time, in his scientific works he does not discuss philosophical issues, but speaks about the development of ideas and the complexity of research, for example: "The experimentalist must know exactly what he has done to the living tissue before he can interpret his experiment. In the past it too often happened that a physiologist would stab into a living frog, take his kymograph records, and then throw the carcass into the waste-jar. This is no longer regarded as good physiology. Without the guidance of accurate anatomical knowledge, sound physiology is impossible; and, without skillful physiological experimentation, the anatomical facts are just facts and nothing more.» But, as Thomas Henry Huxley pointed out, beautiful hypotheses are destroyed by nasty, ugly facts. Facts are powerful both in creating and negating hypotheses.

People tend to perceive information that corresponds to their formed paradigm - beliefs, and ignore the opposite. This is confirmation bias: it leads to polarization, distorted perception of reality and erroneous decisions. Confirmation bias is one of the most common cognitive distortions. See response to the following comment. This is true.

In science, facts are a realized experiment, experience and repeatability (experiment) criterion. We were initially puzzled by this statement. When we document a frog eating a fly, it’s a fact that the frog ate the fly, it’s not so much a realized experiment. But we now understand that to be science, you feel a fact should come from an experimental result, whereas we feel (in the sense of Thomas Henry Huxley and Charles Darwin) that a fact can come from anywhere, and some of the most powerful facts (the ones that destroy hypotheses) often come from animals behaving in nature.

Observational facts are a staple of field-based ecology. See, for example, Sagarin and Pauchard’s 2012 “Observation and Ecology,” Island Press and Real and Brown’s 1991 “Foundations of Ecology” U. Chicago Press)

The scientific process as generally understood proceeds from observation (field or lab) to hypothesis, to hypothesis testing (including experimentation), to theory. We understand that you work at the back end of this process (hypothesis testing through experimentation, and theory formation) while we work at the front end (observations and hypothesis formation). Both ends of this spectrum by themselves are legitimate scientific pursuits. However, hypothesis testing and theory formation depend on initial observations (often unacknowledged) leading to hypothesis formation, and therefore cover the range of scientific endeavor. In contrast, observations and hypothesis formation are on the front end, at the beginning of the scientific process. If these observations are solid (in our case, they are primarily from peer-reviewed journals) the hypotheses formed from them provide value by leading the way towards discovery. For example, Einstein’s four Annus mirabilis papers plus his general relativity paper represent hypotheses based on (mathematical) observations. These hypotheses were later tested by others with sophisticated experimental apparati. Nobody has even accused Einstein of not being a scientist.

P.S.

" It is interesting that the same author (Herrick ) wrote a book: "Fatalism or Freedom: A Biologist's Answer" (Herrick, 1926): "This excursion by a biologist into a field usually posted "No Trespassing" from this side has a two-fold purpose: first, to inquire how far it is profitable to push the inquiry into the problems of human behaviour with the ordinary methods of natural science and so to knit our conscious and social life in with the rest of our world of experience in lawful fashion; and second, to see whether the conclusions thus reached shed any light upon the most acute problems of human conduct self-control, self-determination, self-culture, social control, personal and social morality." It is.